# Eomesodermin in conjunction with the BAF complex promotes expansion and invasion of the trophectoderm lineage

Alexandra Maria Bisia ⑩ , Maria-Eleni Xypolita, Elizabeth K. Bikoff, Elizabeth J. Robertson ⑩ ✉ & Ita Costello ⑩

The T-box transcription factor (TF) Eomesodermin/Tbr2 (Eomes) is essential for maintenance of the trophectoderm (TE) lineage, but the molecular mechanisms underlying this critical role remain obscure. Here, we show in trophoblast stem cells (TSCs) that Eomes partners with several TE-specific TFs as well as chromatin remodellers, including Brg1 and other subunits of the BAF complex. Degron-mediated Eomes protein depletion results in genome-wide loss of chromatin accessibility at TSC-specific loci. These overlap with a subset of sites that lose accessibility following Brg1 inhibition, suggesting that Eomes acts as a "doorstop" controlling TSC chromatin accessibility. Eomes depletion also causes transcriptional misregulation of TSC maintenance and early differentiation markers. An additional subset of Eomes-dependent genes encode intercellular/matricellular interaction and cytoskeletal components, likely explaining the implantation defects of Eomes-null embryos. Thus, Eomes promotes TE lineage maintenance by sustaining trophectoderm-specific chromatin accessibility, while promoting the gene regulatory networks that modulate expansion and cell behaviour during implantation.

The first lineage decision during early mouse development occurs 2.5 days after fertilisation (embryonic day 2.5, E2.5) at the morula stage with the segregation of the outermost trophectoderm (TE) from the inner cells which adopt inner cell mass (ICM) fate. Both lineages can be isolated and indefinitely passaged in vitro in the form of trophoblast stem cells (TSCs) and embryonic stem cells (ESCs) respectively[1–3], providing widely used tractable systems for investigating the genetic pathways and key transcriptional regulators underlying self-renewal and cell fate commitment during early embryogenesis.

The TE lineage plays two essential sequential roles. At the blastocyst stage, the mural TE, which envelops the blastocoel, is responsible for mediating implantation of the embryo via attachment and invasion into the uterine epithelium (reviewed in ref. 4). By contrast, the polar TE population overlying the ICM proliferates to form the so-termed extra-embryonic ectoderm (ExE) that contains the stem cell populations that subsequently diversify to form the progenitor subtypes of the spongiotrophoblast and labyrinth of the mature placenta.

Survival and expansion of the ExE is known to be dependent on the provision of FGF and Nodal signals provided by the underlying epiblast[5,6]. TSCs can be isolated from both pre- (E3.5) and post-implantation (E6.5) embryos[3,7] and maintained in a self-renewing state in the presence of FGF4 and TGF-β[3,8]. Furthermore, when re-introduced into host blastocysts and transferred into pseudopregnant females they contribute to all the diverse trophoblast subtypes of the mature placenta[3].

We and others have identified the T-box gene Eomesodermin (Eomes) as a key TF in the TE lineage[9–11]. Eomes is first induced throughout the TE at the early blastocyst stage, then robustly expressed in the ExE following implantation, and finally becoming confined to the proximal ExE prior to gastrulation, where the multi-potent TE stem cells reside[10,12]. Eomes-null embryos fail during peri-implantation, a phenotype mainly attributed to the lack of expansion of the polar trophoderm. Although Eomes-null blastocysts implant, they do not develop and arrest around E5.0, with embryos displaying

Sir William Dunn School of Pathology, University of Oxford, Oxford, UK. ✉e-mail: elizabeth.robertson@path.ox.ac.uk

an absent ExE and reduced epiblast[10,13]. Moreover, they lack the ability to attach and form blastocyst outgrowths in vitro[9,10]. However, the exact Eomes requirements underlying developmental failure have yet to be determined. Eomes is robustly expressed in TSCs where it has been shown to act together with Elf5 and Tfap2c to regulate the expression of stem cell genes[14]. siRNA-mediated knock-down of Eomes in TSCs results in loss of self-renewal markers including *Elf5* and *Fgfr2*, and the acquisition of a differentiation phenotype[15]. Although Eomes expression by itself is not sufficient to reprogramme cells to a full TS-like state, forced expression of Eomes in conjunction with additional transcription factors, such as Tfap2c, Gata3, Ets2, and Esrrb, is sufficient to drive fibroblasts to a TS-like state[16–18].

Here, we have exploited TSCs to further investigate the molecular mechanisms by which endogenous Eomes functions in the TE lineage. We used rapid immunoprecipitation and mass spectrometry of endogenous protein (RIME) in combination with Cleavage Under Targets and Release Using Nuclease (CUT&RUN) approaches to identify Eomes-interacting protein partners and co-regulated target genes. Interestingly, in addition to a suite of TFs with known roles in TSC maintenance, our experiments also show that endogenous Eomes interacts with a number of chromatin remodelling enzymes including components of a mammalian SWI/SNF complex (canonical BAF, cBAF), known to be critical in establishing and maintaining the TE lineage during both pre- and peri-implantation development[15,19–21].

Additionally, we exploited TSCs carrying an Eomes-degron allele[13] that allows for acute Eomes protein depletion. This allowed us to assess the immediate effects on chromatin accessibility and transcriptional output. We show that Eomes depletion and inhibition of the interacting cBAF chromatin remodelling complex activity both result in loss of chromatin accessibility across the genome. RNA-seq analysis shows that loss of Eomes rapidly leads to down-regulation of TSC markers. Moreover, a subset of directly Eomes-regulated target genes include a number of cytoskeletal, cell adhesion and ECM components consistent with a critical role for Eomes during implantation. Collectively, our experiments provide new insights into how Eomes functions both in the early TE to mediate invasion into the uterine environment, as well as working collaboratively with TSC-specific TFs and chromatin remodellers to suppress early differentiation genes and promote maintenance of the ExE stem cell population that contains the progenitors of the mature placenta.

## Results
### Eomes protein interacts with TSC-specific TFs and chromatin remodelling components, with co-occupancy at TSC-related genes

Eomes function is essential at multiple stages in diverse tissues throughout mouse embryonic development[9,10,22–26]. Loss-of-function mutants arrest at implantation due to Eomes requirements in the TE lineage[10,11]. Here, we sought to identify components enabling Eomes to carry out its TE-specific functions. We carried out RIME[27] to identify Eomes-interacting proteins in wild-type TSCs (Fig. 1a, b). Among Eomes interactors we identified a number of TFs known to be essential for TE/trophoblast (TB) development or function, including Esrrb[28], Zfp281[29], Pou3f1 (Oct6)[30,31], and Klf5[32], as well as the previously-identified interactor Tfap2c (Ap2γ)[14,33,34] (Fig. 1a, b). Furthermore, we identified numerous chromatin regulators including Kdm1a (Lsd1), previously shown to be required for ExE development and differentiation potential of TSCs[35–37] in addition to components of the Nucleosome Remodelling and Deacetylase complex (NuRD), including Chd4 which is critically required for TE specification[38] (Fig. 1a, b; Supplementary Data 1 and 2). We also observed several subunits of the highly conserved multi-subunit cBAF (canonical Brg1/Brm associated factor) complex, a SWI/SNF chromatin remodelling complex. These include the catalytic ATPase subunit Brg1 (Smarca4), in addition to Smarcc1, Smarcd1, Smarcb1, Smarcd2 and the cBAF specific subunit Arid1a (Fig. 1a, b). The BAF complex components have known roles in pre- and peri-implantation development and in the trophectoderm lineage[19–21]. Overall, the TSC-specific Eomes ineractome has critical roles in TE lineage and implantation development.

Next, we performed CUT&RUN-seq experiments in TSCs to identify Eomes chromatin binding sites. This allowed us to identify 4145 sites, mostly located distally from promoters (81.4% within 5–500 kb of the transcriptional start site, TSS) consistent with previous work suggesting Eomes occupies distal regulatory elements[25] (Fig. 1c). These Eomes-bound sites are enriched for consensus binding motifs recognised by TFs expressed in TE in vivo and TSCs in vitro (Fig. 1d), including those shown by RIME to interact with Eomes (Fig. 1a, b). GO analysis of the nearest genes ("Eomes-bound genes") showed enrichment for terms related to the TE lineage and TGF-β signalling, known to be essential for TSC self-renewal (Fig. 1e, Supplementary Data 3). Moreover, using previously published ChIP-seq data sets[28–30], we found a high percentage of co-occupancy by one or more of its known interaction partners (Fig. 1f), including at TSC maintenance (Elf5, Fgfr2) and early differentiation genes (Hand1, Cited2)[39–43], as well as genes encoding components of cell-cell and cell-extracellular matrix (ECM) interactions (Lamb1, Cdh1) (Fig. 1g–i). These data strongly suggest that Eomes co-regulates TSC-specific genes by directly interacting with other TFs of the TSC gene regulatory network (GRN), and co-binding to chromatin regulatory regions. Interestingly, we found Eomes binds ~90 kb upstream of its own TSS, at a site previously identified as a superenhancer region that directly interacts with the Eomes promoter region[30] (Fig. 1g).

### Acute Eomes depletion initiates differentiation in Eomes[deg/deg] TSCs

To investigate the functional roles of Eomes in TSCs we took advantage of the previously described mouse line harbouring a degron-tagged Eomes allele[13] to derive homozygous Eomes[deg/deg] TSCs, enabling us to rapidly and completely deplete Eomes protein. Immunofluorescence staining showed that Eomes[deg/deg] cells express Eomes and Tfap2c comparably to wild type (WT) TSCs (Supplementary Fig. S1a). As expected, dTAG-13 addition resulted in rapid and sustained total protein depletion, with Eomes at undetectable levels after 1 hour (h) by immunoblotting (Supplementary Fig. S1b). Longer-term Eomes depletion also resulted in reduction of Fgfr1 (Supplementary Fig. S1c), which is necessary for TSC maintenance[44], and more intense F-actin phalloidin staining at the periphery of TSC colonies[45,46] (Supplementary Fig. S1d). Concomitantly, there was an increase in the proportion of cells expressing Hand1 (Supplementary Fig. S1e), an early marker of TSC differentiation[42,43,47]. These results strongly suggest that acute, sustained Eomes depletion initiates a differentiation trajectory in Eomes[deg/deg] TSCs.

### Eomes depletion and Brg1 inhibition both result in loss of chromatin accessibility across the genome

Brg1 (Smarca4) has been shown to govern genome-wide chromatin remodelling and its ATPase-dependent chromatin remodelling activity is critical for the function of the BAF complex[48]. Interestingly, similar to Eomes, Brg1 is essential for early development. Brg1-null blastocysts are pre-implantation lethal, fail to hatch from the zona pellucida or implant into the uterus. They also fail to form outgrowths in ex vivo cultures[19]. To investigate the functional relevance of Eomes-Brg1 interactions in TSCs, we next carried out an Eomes depletion time-course in Eomes[deg/deg] TSCs, alongside a Brg1 inhibition ("BRGi") time-course following the addition of the Brg1 small molecule ATPase inhibitor BRM014[49] to the culture medium. BRM014 inhibits the catalytic activity of both Brg1 and Brm (Smarca2), the two ATPase subunits that characterise BAF complexes. As Brg1 is the main SWI/SNF ATPase expressed in trophectoderm, we refer to it here as a Brg1-inhibitor[19,50]. Transcriptional and chromatin accessibility changes were then

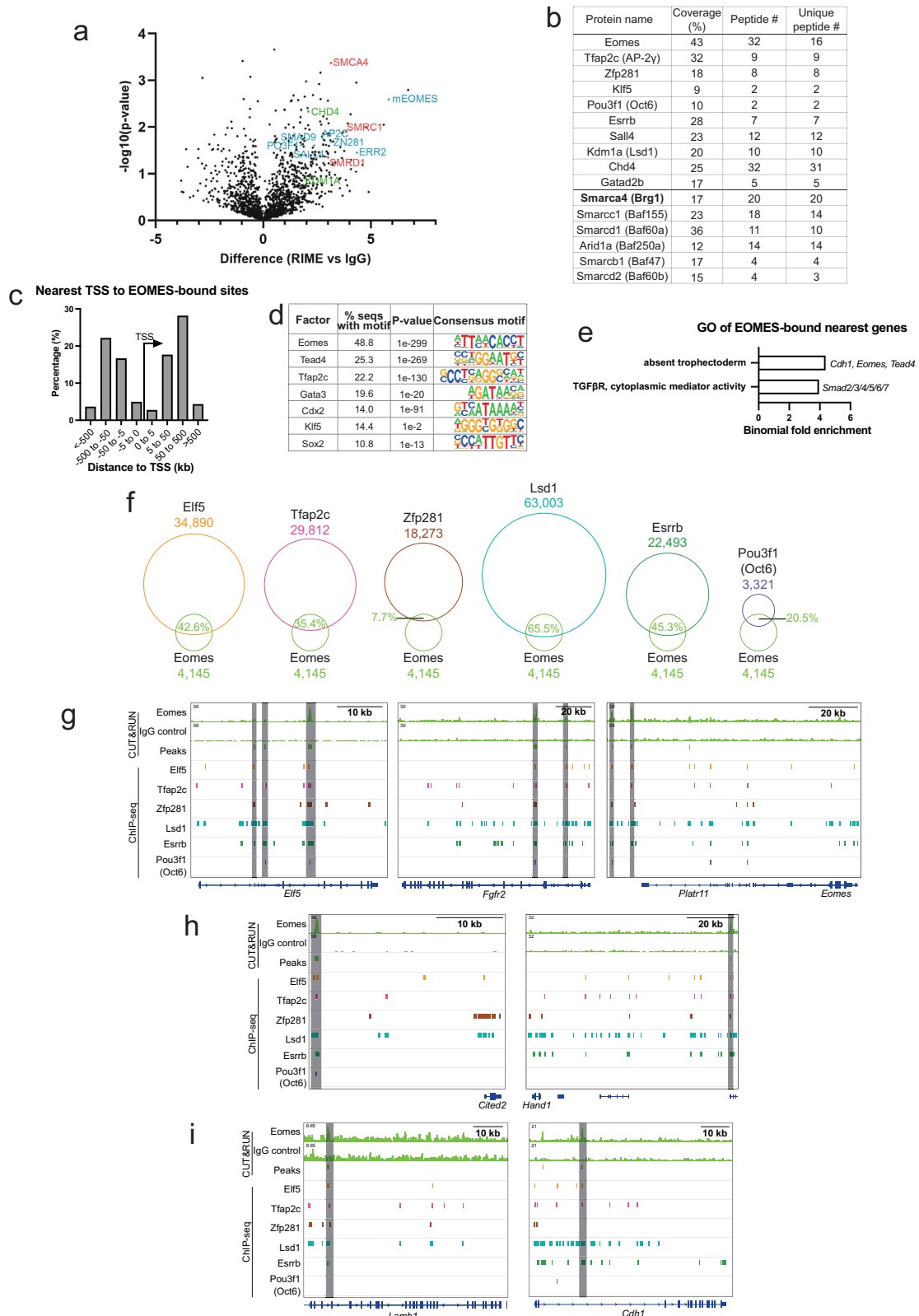

**Fig. 1 | Eomes interacts with TSC-specific TFs and shows co-occupancy at TSC maintenance and differentiation genes. a** Volcano plot of Eomes interactors identified by RIME, generated by a two-sided t-test. Selected interactors are highlighted, TF's are indicated in blue, BAF components in red and other chromatin modifiers in green. **b** RIME summary for all samples combined for selected Eomes interactors. **c** Bar graph of distance distribution of Eomes-bound regions identified by CUT&RUN-seq from nearest transcriptional start sites (TSS). **d** Highlighted motif enrichment analysis in Eomes-bound genomic regions. Shown are HOMER binomial *P* values. **e** Enriched gene ontology (GO) terms analysis of Eomes-bound genes. **f** Schematic of Eomes chromatin occupancy overlap with previously-published ChIP-seq datasets for other TSC-specific transcription factors. (Latos et al. 2015; Lee et al., 2019). **g**–**i** Genomic tracks of CUT&RUN-seq Eomes and IgG control samples, macs2-called peaks, and previously-published ChIP-seq peaks of TSC-expressed TFs [as in (**f**)]. The selected tracks show TSC markers (**g**), early differentiation markers (**h**), and mediators of cell-cell and cell-ECM interactions (**i**). Highlighted regions indicate sites of TF co-occupancy with Eomes.

examined by bulk ATAC- and RNA-seq. As it has previously been shown in ESCs that BRGi acts within minutes[51], we also included a 1 h treatment timepoint for this condition (Supplementary Fig. S2a). To control for cell density-dependent effects on the data, we also included a time course of DMSO-treated control cultures (Supplementary Fig. S2a).

Peak calling analysis of the ATAC-seq dataset indicated that Eomes depletion and BRGi both result in statistically significant reduction of the overall number of accessible sites in the genome, for BRGi after 1 h and for Eomes depletion after 48 h (Supplementary Fig. S2b). Differential accessibility analysis largely showed reduction of accessibility at an increasing number of sites from 12 h of Eomes depletion onwards (Fig. 2a, Supplementary Fig. S2c). These less-accessible sites mostly represent distal regulatory elements, with the majority situated between 5 and 500 kb from the nearest TSS (Fig. 2b). We overlapped all differentially-accessible sites with previously-published histone acetylation (H3K27ac) and monomethylation (H3K4me1) ChIP-seq datasets[30]. These histone modifications mark active and active/poised enhancers, respectively[52,53]. 48 h after Eomes depletion, we observed significant overlap between these datasets, suggesting that regions whose accessibility is affected upon Eomes loss represent enhancer regions (Supplementary Fig. S3a, b, e). Gene ontology (GO) analysis of the genes nearest Eomes-depleted differentially-accessible sites showed enrichment for TE or TE-derived cell abnormalities (Fig. 2c), consistent with the role of Eomes in the TE development[9,10]. Interestingly, we also found terms related to cell interactions with, and migration through, their environment (Fig. 2c). Later timepoints showed enrichment for similar terms (Supplementary Fig. S2d, e). Finally, less-accessible sites upon Eomes depletion are enriched for consensus binding motifs of TSC-expressed TFs, including Eomes interactors (such as Tfap2c, Pou3f1) (Fig. 2d, Supplementary Fig. S2f). Eomes may thus be binding these sites in complexes containing other TSC-specific TFs.

Differential accessibility analysis similarly indicated that BRGi broadly results in reduced accessibility at thousands of sites across the genome as early as 1 h after treatment (Fig. 2e, Supplementary Fig. S2b). These include sites near TSC-related genes including Eomes and Cited2. As with Eomes depletion, sites showing reduced accessibility upon BRGi are mostly located distal to TSSs, between 5 and 500 kb away from TSSs (Fig. 2f, Supplementary Fig. S2g). Similarly with Eomes depletion, regions with BRGi-induced differential accessibility are occupied by H3K27ac and H3K4me1 marks, indeed to a greater degree (Supplementary Fig. S3c–e), again suggesting the affected regions represent enhancers. BRGi sites of reduced accessibility are also enriched for the consensus motifs of several TSC-expressed TFs, including Eomes itself (Fig. 2g). Taken together, these results suggest that the SWI/SNF complex broadly acts to promote accessibility at distal regulatory elements in TSCs.

To examine the possible relationship between differentially-accessible sites associated with Eomes depletion or BRGi respectively, we overlapped the differentially-accessible sites at corresponding treatment timepoints. At each timepoint, the vast majority of differentially accessible sites arising as a consequence of Eomes depletion overlap with differentially accessible sites resulting from BRGi treatment (Fig. 2h). Thus in TSCs, Eomes in conjunction with Brg1 appears to be playing a "doorstop" role in the sustained maintenance of accessibility of TE- and TB-associated loci of the genome. Once the Eomes doorstop is removed or the Brg1 catalytic activity is blocked, chromatin regions lose accessibility, preventing access by other TFs to the distal regulatory elements. Hence, the Eomes/Brg1 interaction provides a sensitive switch for the regulation of these regulatory elements.

## Eomes depletion results in mis-regulated expression of TSC maintenance and differentiation genes, as well as cytoskeleton, matricellular and cell-cell interaction genes

Next, we examined the changes in gene expression that occur following acute Eomes depletion. As early as 6 h after dTAG-13 addition

we already find a sub-set of genes become differentially expressed (Fig. 3a, Supplementary Data 4). Although there are similar numbers of genes up- and downregulated, when using a cutoff of ≥1.5-fold, we find the majority of genes are those that lose expression as a consequence of Eomes loss (Fig. 3b). Principal component analysis (PCA) shows that Eomes-depleted TSCs follow a different transcriptional trajectory versus DMSO treated control cultures (Fig. 3c). Notably, 6 h after dTAG-13 addition, TSCs are already distinct from untreated controls (Fig. 3c). Thus, Eomes loss has a rapid, detectable effect on the transcriptome as early as 6 h after treatment. Established markers of TSC identity and maintenance are gradually downregulated, including Elf5, Fgfr1, Cdx2, and Esrrb. Tfap2c, which is essential for TSC maintenance but also for priming cells to express differentiation markers[14] is initially upregulated, before becoming slightly downregulated after 48 h. Notably, Eomes transcripts are upregulated following Eomes protein depletion, suggesting the presence of an auto-regulatory loop. Indeed, in addition to Eomes, Tfap2c, Gata3 and Ets2 are TE genes that are also initially upregulated at 6 h after Eomes depletion. These 4 TFs have previously been shown to jointly reprogramme fibroblasts to TS-like cells[16], and hence may represent an auto-regulatory TF network required for TSC maintenance in self-renewal culture conditions. Genes that are expressed in the ExE, such as Elf5 and Sox2, show a clearer downregulation over time. In contrast, early markers of differentiation, including Hand1 and Cited2 are rapidly upregulated from the first 6 h of treatment (Fig. 3d). Thus, Eomes acts both as an activator promoting expression of TSC GRN components, as well as a repressor of TSC differentiation and early TB markers.

We further overlapped the list of differentially-expressed genes with those identified by Eomes CUT&RUN-seq to identify those representing directly Eomes-regulated genes (Fig. 3e, f). As early as 6 h after depletion, GO analysis of directly regulated genes included terms related to the development and differentiation of the TE and TE-derived tissues (Fig. 3g, h). 48 h post-treatment, directly-regulated downregulated genes are enriched for terms related to the FGFR and TGFβR signalling pathways, both essential for TSC maintenance (Supplementary Fig. S4a). Meanwhile, directly-regulated genes more robustly expressed in Eomes-depleted TSCs are enriched for terms related to TE differentiation, such as 'cell differentiation involved in embryonic placenta development,' 'labyrinthine layer development' (Supplementary Fig. S4b). Interestingly, there was an enrichment for terms related to cell-cell interactions, cell motility, and cytoskeletal organisation, both at early (Fig. 3g, h) and later (Supplementary Fig. S4a, b) timepoints.

Thus, Eomes seems to directly upregulate genes associated with TSC/TE maintenance (Fgfr1, Sox2, Elf5, Zfpm1, Id2); and downregulate genes controlling early TSC/TE differentiation (Cited2, Hand1, Tfap2c). Interestingly, Eomes also directly regulates genes only recently identified in the TSC context (Meis1, Pou3f1, Zfpm1, Klf6)[30,54–57] (Fig. 3e, Supplementary Fig. S4c). Furthermore, Eomes also directly regulates a variety of genes associated with the cytoskeleton (Actg1), cell-ECM adhesion (Lama1, Lama3), cell-cell adhesion (Cdh1), and ECM components (Col4a1, Col7a1)[58] (Fig. 3e, Supplementary Fig. S4c).

## Eomes depletion and BRGi result in concordant gene misregulation

As Eomes depletion and BRGi result in largely overlapping losses in chromatin accessibility, we additionally examined the effect of BRGi on gene expression. As soon as 1 h after BRGi changes in gene expression become evident (Fig. 4a). Following 48 h of inhibition, GO analysis identifies enriched terms related to TE and TE-derived cell types, suggesting Brg1 specifically contributes to TSC context-specific gene regulation (Fig. 4b). When we overlapped Eomes depletion- and BRGi-misregulated genes, we found that the majority (74.4%) of the former are included in the latter (Fig. 4c). Finally, a dot plot of the log2-fold changes of misregulated genes

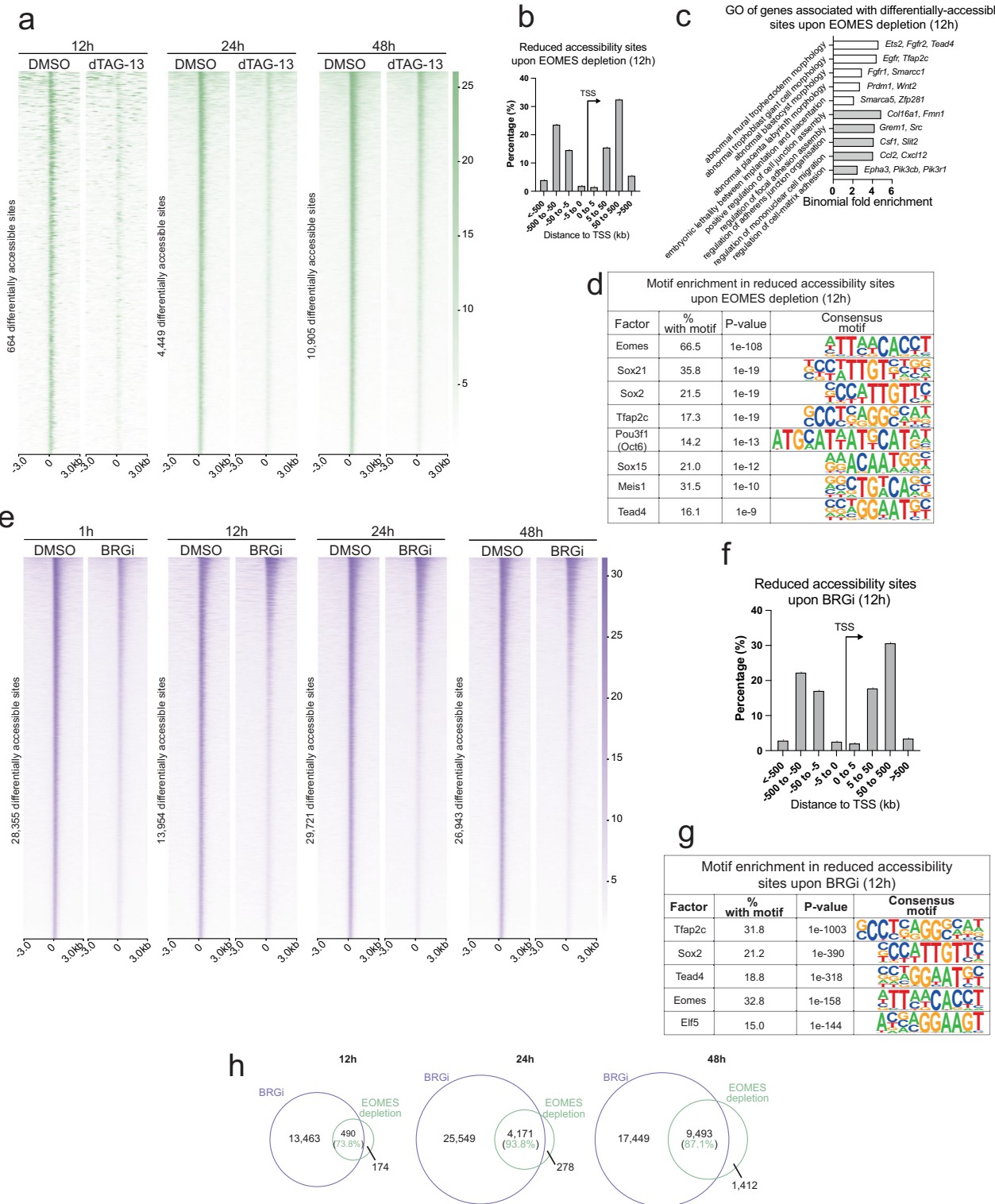

**Fig. 2 | Both Eomes depletion and Brg1 inhibition ("BRGi") result in loss of accessibility at distal regulatory regions. a** Heatmaps of differentially-accessible sites at 12, 24, and 48 h after Eomes depletion. **b** Distance distribution of sites showing reduced accessibility 12 h after Eomes depletion from nearest TSS. **c** Selected GO terms of genes nearest differentially-accessible sites 12 h after Eomes depletion. White bars correspond to TE-related terms, grey bars correspond to cell interactions and motility-related terms. **d** Enriched consensus motifs at sites of reduced accessibility 12 h after Eomes depletion. Shown are HOMER binomial *P* values. **e** Heatmaps of differentially-accessible sites at 1, 12, 24, and 48 h after BRGi. **f** Distribution of distances from nearest TSS of sites with reduced accessibility 12 h after BRGi. **g** Selected enriched consensus motifs at sites of reduced accessibility after 12 h of BRGi. Shown are HOMER binomial *P* values. **h** Schematic of overlap between differentially accessible sites at 12, 24, and 48 h timepoints between Eomes-depleted and Brg1-inhibited TSCs.

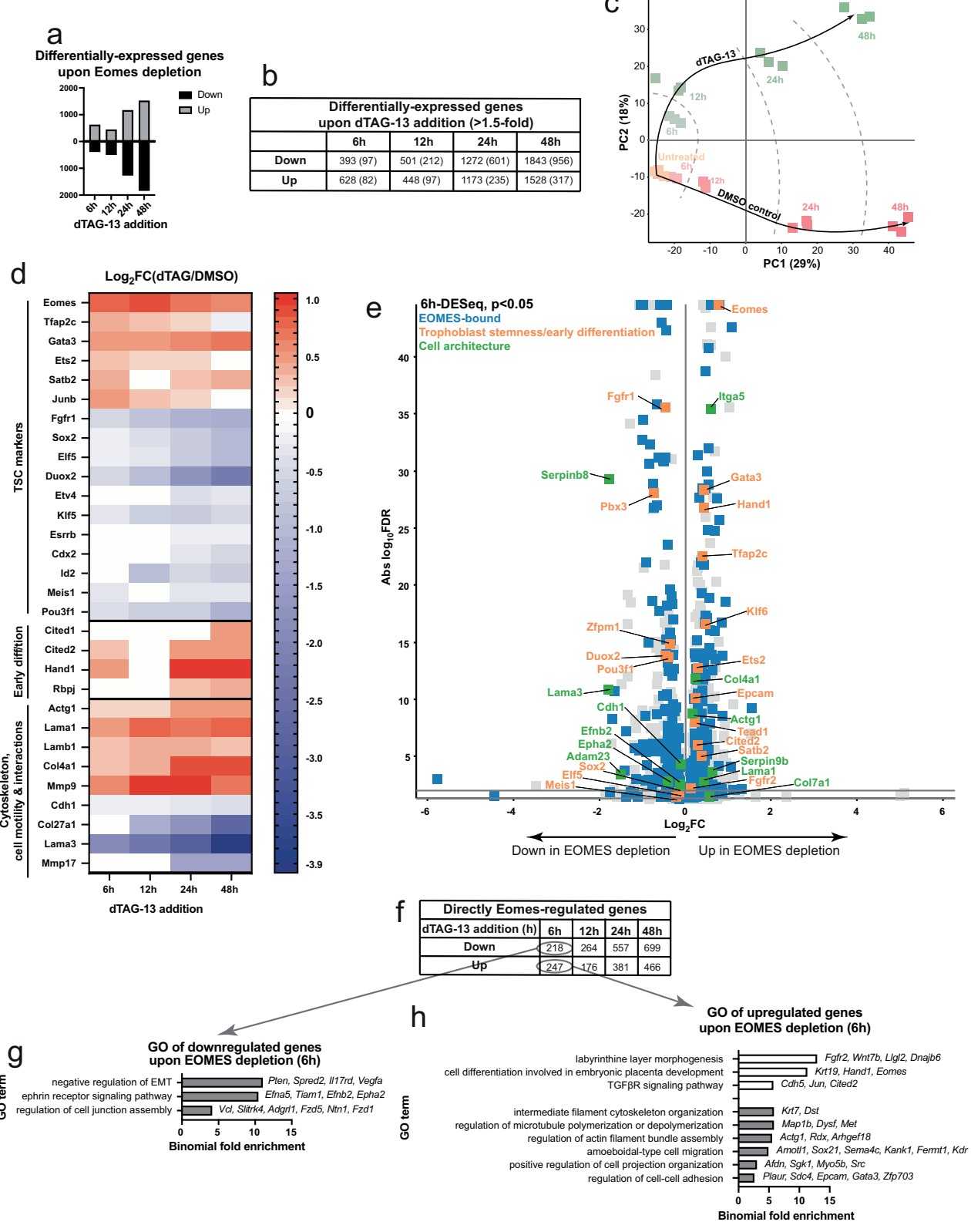

**a** Differentially-expressed genes upon Eomes depletion

**b**

| Differentially-expressed genes upon dTAG-13 addition (>1.5-fold) | | | | |
|---|---|---|---|---|
| | **6h** | **12h** | **24h** | **48h** |
| **Down** | 393 (97) | 501 (212) | 1272 (601) | 1843 (956) |
| **Up** | 628 (82) | 448 (97) | 1173 (235) | 1528 (317) |

**c**

**d** Log₂FC(dTAG/DMSO)

**e** 6h-DESeq, p<0.05
EOMES-bound
Trophoblast stemness/early differentiation
Cell architecture

**f**

| Directly Eomes-regulated genes | | | | |
|---|---|---|---|---|
| dTAG-13 addition (h) | **6h** | **12h** | **24h** | **48h** |
| **Down** | 218 | 264 | 557 | 699 |
| **Up** | 247 | 176 | 381 | 466 |

**g** GO of downregulated genes upon EOMES depletion (6h)

**h** GO of upregulated genes upon EOMES depletion (6h)

shared between Eomes depletion and BRGi showed a positive correlation, with the majority similarly upregulated or downregulated under both conditions (Fig. 4c). Notably, the log2-fold changes of several members of the TSC GRN (*Eomes, Esrrb, Elf5*), other TSC markers (*NrOb1, Duox2*), early differentiation markers (*Hand1, Cited2*) and components of cellular architecture and cell interactions (*Actb, Col27a1, Serpinb8*) are positively correlated, indicating

concordant regulation of these genes by Eomes and Brg1 activity (Fig. 4d).

**Eomes functions in the mural trophectoderm are essential for successful implantation**

Depletion of Eomes in TSCs results in the deregulation of a sub-set of genes associated with cell-cell and cell-ECM adhesion (Fig. 2c). We

**Fig. 3 | Eomes depletion results in misregulation of TSC-expressed genes.** Bar graph (**a**) and table summary (**b**) of differentially-expressed genes between Eomes-depleted and control TSCs. **c** RNA-seq principal component analysis (PCA) scatterplot of Eomes-depleted and DMSO-treated control TSC transcriptomes. **d** RNA expression log2-fold change heatmap of selected TSC maintenance and differentiation markers, as well as cytoskeleton, cell-cell and matricellular interaction components. **e** Volcano plot produced by DESeq2 of differentially-expressed genes (*P* < 0.05) between Eomes-depleted and control TSCs at 6 h. (grey and all other colours). Cell architecture genes (green) include cytoskeletal subunits, cell-cell and matricellular interaction components. Both indicated trophoblast stemness and early differentiation genes (orange) and cell architecture genes (green) are a subset of Eomes-bound (blue) genes. Full differentially expressed gene list is found in file Supplemental Data 4. **f** Table summary of genes 'directly-regulated' by Eomes (differentially-expressed upon Eomes depletion, as well as Eomes-bound, see Fig. 1). GO terms enriched in directly-regulated genes with reduced (**g**) or increased (**h**) expression 6 h after Eomes depletion.

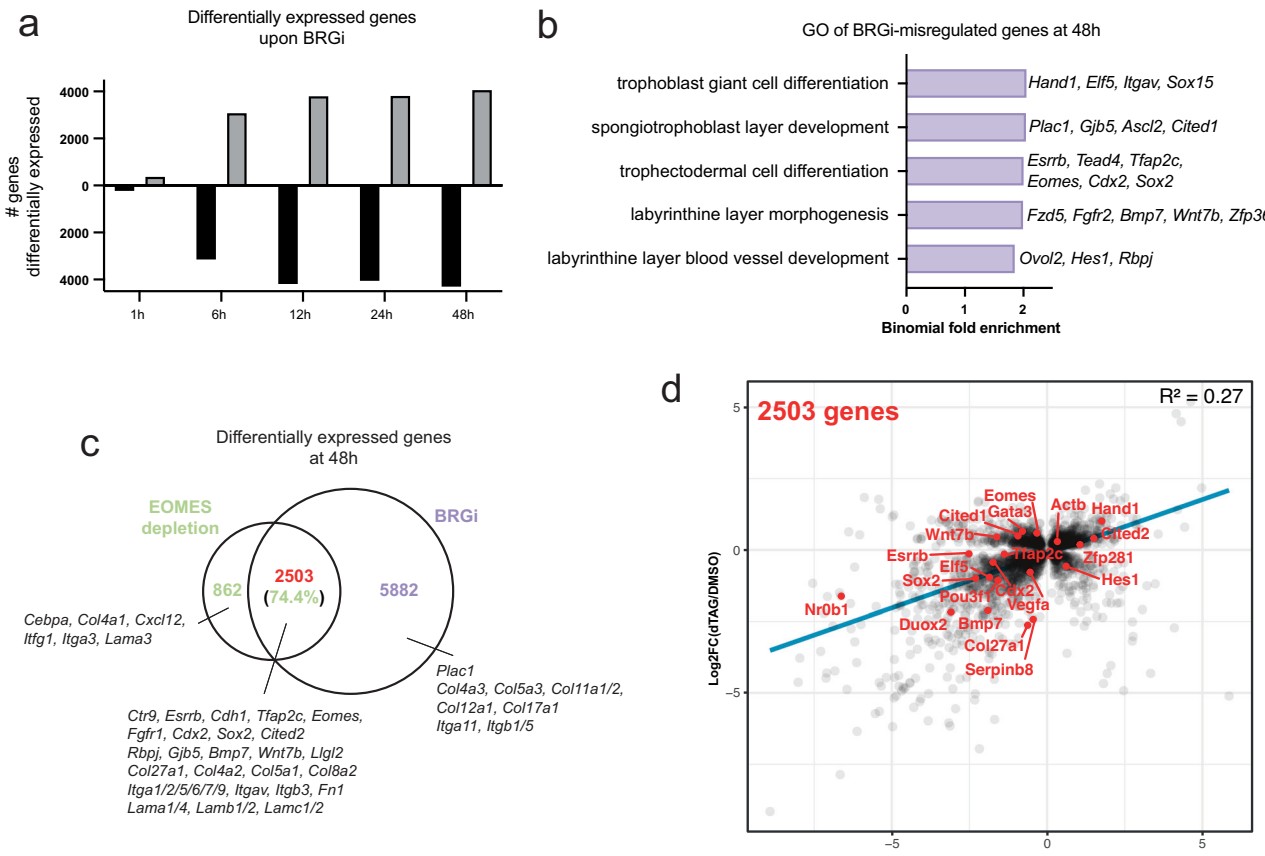

**Fig. 4 | Brg1 regulates expression of TSC-related genes. a** Bar graph of up- and down-regulated genes at each BRGi timepoint. **b** Selected enriched GO terms of genes differentially expressed after 48 h of BRGi. **c** Schematic of overlap between genes differentially expressed after 48 h of Eomes depletion or BRGi. **d** Scatterplot of log2-fold changes of common differentially-expressed genes at 48 h between Eomes depletion and BRGi (**c**). A linear regression line has been fitted.

considered whether this finding reflects in part the central role Eomes plays in the mural trophectoderm. To date, investigation of Eomes in the peri-implantation embryo has focused on its essential requirements in the polar trophectoderm for establishment of the ExE[10]. Notably, however, Eomes-null blastocysts are incapable of attaching and forming outgrowths in culture[9,10]. Moreover, unlike Cdx2 which is abruptly down-regulated[59], Eomes expression in the mural trophectoderm is maintained during implantation. The mural trophectoderm gives rise to the primary trophoblast giant cells (p-TGCs) that attach to and migrate through the maternal luminal epithelium of the implantation crypt and invade the underlying stroma of the primary decidual zone (PDZ). Interestingly, Eomes expression is strongly retained in the invading p-TGCs present in the PDZ[60]. To test whether Eomes is required for successful attachment and invasion in vivo, next we examined the behaviour of the mural trophectoderm in Eomes null embryos. To visualise the embryonic cell population we made use of a paternally inherited ubiquitously expressed Rosa26-membraneTomato knock in allele[61] crossed into the background of the Eomes null mouse strain[9].

Confocal imaging of vibratome sections collected at E5.5 clearly show attachment and invasion of mTomato+ p-TGCs in control Eomes+ embryos (Fig. 5a). In contrast, the mural trophectoderm of Eomes-null embryos fail to transverse the luminal epithelial layer and the embryo remains confined within the uterine crypts (Fig. 5b, c). Thus we conclude that in addition to its role in mediating expansion of the polar trophectoderm, Eomes functions in the mural trophectoderm cell population are essential for invasion through the luminal epithelial cell barrier and subsequent remodelling of the adjacent deciduum.

## Discussion

Eomes has multiple distinct, essential functions in the pre- and post-implantation mouse embryo[9,10,24–26]. This single TF controls the development of several distinct lineages, including cardiac mesoderm, definitive endoderm and yolk-sac fated extraembryonic mesoderm. Here, we focussed on its earliest role in the TE lineage. In vivo and in vitro studies[9,10,15] have previously described its essential functions during implantation and development of the early ExE population. We

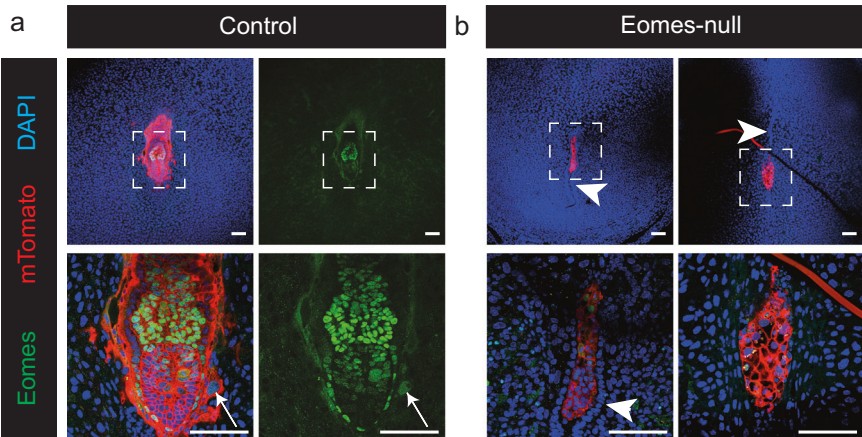

**Fig. 5 | Trophectoderm of Eomes null embryos does not implant and migrate through the uterine epithelium.** Confocal imaging of representative sections of **a** control Eomes-expressing and **b** Eomes-null embryos at 5.5 dpc within the decidua carrying the ubiquitiously expressed transgene mTomato, stained for Eomes and mTomato, counterstained with DAPI. 18 control (Eomes positive) and 4 Eomes-null embryos in the decidua were analysed for this study. Lower panel is a higher magnification image of the embryos in the upper panel. Scale bar = 100 μm. Arrow indicates invaded P-TGC. Arrowhead shows intact uterine epithelium.

exploited a combination of proteomic and genome-wide approaches to further characterise the molecular basis of Eomes function in the TE lineage.

Using RIME we identified the endogenous Eomes interactome in TSCs. The present findings strengthen a previous report showing that the interaction with Tfap2c acts as a switch between promoting TSC maintenance versus differentiation[14]. We also describe Eomes interactions with numerous other TFs that have roles in the TB lineage, including Esrrb, Klf5, Pou3f1, and Zfp281. We also found interactions with numerous chromatin regulators, including Lsd1, the NuRD complex, and the BAF complex, known to play essential roles in TB development. Comparison of Eomes-bound regions with regions bound by interacting TFs and chromatin regulators show a varying degree of overlap across the genome at distal regulatory elements. Thus, Eomes does not form a homogeneous complex, instead interacting with various TFs dynamically at different sites across the genome to exert its action. Intriguingly, Esrrb has also been shown to interact with Zfp281, Junb, Lsd1 and the NuRD complex in TSCs[28]. Additionally, Zfp281 has been shown to interact with both promoters and enhancer in TSCs and these binding sites also show enrichment for Tfap2c, Lsd1 and Esrrb[29]. Collectively, these results reveal the complex multifaceted functional role played by Eomes and its associated interacting proteins in TE development, while illustrating how a single TF can have multiple roles in different cellular or developmental contexts.

Chromatin structure regulates TF binding, mainly by modulating chromatin accessibility, thus determining cell type-specific gene expression programmes. Our RIME experiment revealed that Eomes interacts with numerous subunits of the canonical ATP-dependent BAF (mammalian SWI/SNF) chromatin remodelling complex. The BAF complex, critical at several developmental stages, is thought to exert its action by ATPase-driven sliding of nucleosomes along the DNA, thereby rendering these regions accessible to binding by other factors[62,63]. Importantly, BAF subunits, such as Baf155/Smarcc1, Brg1/Smarca4 and Baf47/Smarcb1 have been shown to be required in pre and/or peri-implantation embryos[19–21]. Furthermore, Brg1 has previously been shown to interact with Eomes during the acquisition of innate memory in CD8 + T cells and the role of Eomes is Brg1-dependent at numerous target genes[64]. More recently, Eomes was also shown to interact with a SWI/SNF complex, similarly binding to mesodermal and endodermal gene enhancers in an in vitro differentiation model of mesoderm and endoderm specification[65]. The cBAF complex has also been shown to play a role in human TB

development[66]. The interaction of Eomes with the cBAF complex hence reveals one of the molecular mechanisms through which Eomes exerts its role in TSCs, and Eomes interaction with SWI/SNF complexes may be a common theme in its molecular function.

It has previously been shown that continuous BAF complex function, and the Brg1 subunit specifically, is essential to maintain chromatin accessibility and appropriate gene expression[48,67,68]. Here, we exploited acute Eomes depletion through the dTAG system[69] alongside small-molecule inhibition of Brg1, to probe the functional relationship of these regulatory components in the context of TSCs. Indeed, Eomes depletion or Brg1 inhibition both result in genome-wide reduction in chromatin accessibility. Furthermore, a large majority of Eomes-dependent accessible sites (>70%) are also rendered inaccessible upon Brg1 inhibition. Previously, Eomes has been shown to have a pioneer role upon activation of its expression[70,71]. Additionally, the BAF complex interacts directly with other cell type-specific pioneer TFs, such as Oct4 in ESCs and PU.1 in leukemia cells[72,73]. As Eomes is constitutively expressed in TSCs, its role in this context appears to be acting as a "doorstop"[74,75] guiding Brg1 to sustain BAF-mediated accessibility at TSC-specific sites, as both Eomes and Brg1 are required for sustained accessibility at these sites. These accessible sites may then act as "landing pads" for binding of additional transcriptional regulators to control TSC-specific gene expression.

After implantation, Eomes is expressed in the ExE, a polar TE derivative, which overlies the epiblast and contains the stem cell niche that contributes the progenitors of diverse placental cell types, and which is maintained by a signalling environment of high FGF and Nodal from the epiblast[3,76]. As the ExE expands and cells are displaced proximally from the source of FGF and Nodal signals, the associated loss of Eomes correlates with down-regulation of stem cell maintenance genes and derepression of trophoblast differentiation genes[14,77]. In keeping with the idea that Eomes regulates this temporal and spatial cell fate switch in vivo, here we find that acute Eomes depletion alone in the context of sustained FGF/TGF-β signalling is sufficient for the rapid downregulation of TSC markers (*Elf5, Fgfr1/2, Klf5, Esrrb*) and upregulation of early differentiation markers (*Hand1, Cited1/2*) promoting the acquisition of the various placental progenitor and differentiated cell identities required for appropriate placental morphogenesis. In line with our findings that Eomes and Brg1 promote accessibility of largely overlapping chromatin regions associated with TSC-expressed genes, we also find that there is a positive correlation in the change in gene expression upon Eomes

depletion of Brg1 inhibition. Nevertheless, further to its effects on TSC-specific genes, Brg1 inhibition results in significantly more extensive and rapid changes in gene expression, in line with the fact that Brg1 is involved in a more global regulation of gene expression.

Additionally, we identified a host of "cellular architecture" regulated genes including cytoskeletal components (*Actg1*) and components of cell-cell and matricellular interactions (*Lama1, Lamb1, Col4a1, Cdh1*), many of which are known to be expressed in the mural TE[58]. Eomes-null blastocysts form an outer layer of Cdx2-positive TE cells, can hatch from the zona pellucida and elicit a decidual response. However, in vitro, null blastocysts do not attach and migrate on a simple plastic substrate during outgrowth experiments. Our current findings strongly suggest that Eomes modulates the expression of essential genes driving the adhesion to, and invasion of, the uterine environment by mural TE-derived p-TGCs. Indeed, we fail to detect any embryo-derived cells from Eomes-null embryos that have invaded into the uterine decidium, rather we find that Eomes-null embryos are strictly confined to the uterine luminal epithelial crypts. Interestingly, Eomes expression persists in p-TGCs after stromal invasion[60], consistent with a potential role for Eomes in contributing to their normal invasive phenotype.

The roles Eomes plays during peri-implantation development appear to be context-dependent. Indeed, Eomes and its multiple interaction partners bind to an array of different sites across the TSC genome. While its role may appear contradictory, this is likely due to its distinct roles in the mural versus the polar trophectoderm populations. In wild-type embryos, Cdx2 is normally downregulated in the mural TE allowing the expression of cell-adhesion and migration genes. Forced Cdx2 overexpression in TE represses the differentiation of the mural TE into the invasive p-TGCs[59]. Indeed, Eomes null embryos retain Cdx2 expression throughout the TE layer[11]. In the peri/post-implantation embryo, Eomes promotes the expansion of the TSC containing polar trophectoderm via the positive regulation of a subset of TSC genes required for ExE development while repressing the expression of early differentiation markers of the polar trophectoderm. In vivo/ex vivo investigation of Eomes functions in peri/post-implantation embryos is challenging and to our knowledge, currently unexplored. As Eomes is essential during pre-implantation, deductions about its peri- and post-implantation functions have largely been extrapolated from in vitro TSC-based studies. This aspect of Eomes function clearly warrants further investigation but is technically challenging due to the small size of the peri-implantation embryo and associated implantation site. However, the recent development of robust ex vivo TE invasion/peri-implantation models[78] in conjunction with the Eomes-degron mouse line[13], should, in future, provide the opportunity to further address the temporal and spatial requirements of Eomes ensuring appropriate uterine attachment and invasion.

In summary, this study builds on and expands our understanding of Eomes functions in the TE lineage. A key mechanism of Eomes appears to be in its interaction with the cBAF chromatin remodelling complex, of which both Eomes and Brg1 ATPase activity are required for the "doorstopping" function to maintain open accessible regulatory enhancer regions. Additionally, it participates in the regulation of discrete categories of genes related to the distinct essential roles it plays within the TE lineage, namely in directing successful uterine invasion and implantation of the blastocyst, as well as transiently sustaining the stem cell pool within the ExE that generates the trophoblast sub-types of the emerging placenta to ensure successful development of the embryo within the uterine environment. This study furthers our understanding of how a single TF, Eomes, exerts its function in distinct cell lineages both spatially and temporally during a brief but critical time in early mammalian development.

## Methods

### Mouse strains
All animal procedures were performed at the Sir William Dunn School of Pathology, University of Oxford in accordance with Home Office (UK) guidelines, and authorized by the local Animal Welfare and Ethical Committee. Animals were maintained under a 12 h light/dark cycle and housed in individual ventilated environmentally controlled cages (20 °C, 55% humidity). Sexually mature *Mus musculus* males and females (>6 weeks of age) were intercrossed to generate embryos at indicated ages, e.g., 5.5 days post coitum (dpc). Eomes[deg/deg] mice were genotyped as previously described[13]. Eomes[+/-]:Rosa26[mT/mG] males were mated with Eomes[+/-] females to generate embryos for immunofluorescence imaging. Eomes[+/-] and Rosa[mT/mG] alleles were genotyped as previously described[9,61]. All mice were on a C57/Bl6 background.

### Rapid immunoprecipitation and mass spectrometry of endogenous proteins (RIME) and proteomic analysis
Wild-type TSCs[79] were fixed with 1% methanol-free formaldehyde for 8 min at room temperature (RT) and quenched by incubating in 0.125 M glycine for 5 min. Cells were scraped, collected, washed once with chilled PBS containing 0.5% Igepal then washed with PBS-Igepal containing 1 mM PMSF. Cell pellets were snap-frozen. RIME analysis was performed by Active Motif using previously published protocols[27]. 4 Eomes RIMEs were conducted alongside 2 IgG control RIMEs. Proteins of interest were immunoprecipitated from 150 µg of pre-cleared chromatin with 15 µg of Eomes antibody (Abcam, ab23345) or rabbit IgG antibody control, followed by LC-MS/MS on a Thermo Scientific Q Exactive Orbitrap Mass spectrometer linked to Dionex Ultimate 3000 HPLC (Thermo Scientific) and a nanospray Flex™ ion source. Tandem mass spectra were extracted and analysed by PEAKS Studio version 8 built 20. Charge state deconvolution and deisotoping were not performed. Database consisted of the Uniprot database (version 180508, 71,771 curated entries) and the cRAP database of common laboratory contaminants (www.thegpm.org/crap; 114 entries). Database was searched with a fragment ion mass tolerance of 0.02 Da and a parent ion tolerance of 10 PPM. Peaks studio built-in decoy sequencing and FDR determination was used to validate MS/MS based peptide and the parsimony rules for protein identifications. A threshold of the −10*logp (p-value) of 20 or greater was applied for the peptide identifications. The weighted sum of 9 parameters for peptide scoring were converted to a p-value which represent the probability of a false identification. Protein identifications were accepted if they passed the −10logp of 20 and contained at least 1 identified unique peptide. Proteins that contained similar peptides and could not be differentiated based on MS/MS analysis alone were grouped to satisfy the principles of parsimony. Proteins sharing significant peptide evidence were grouped into protein groups.

### TSC derivation and culture
TSCs from wild-type, Eomes[deg/deg] and Eomes[V5/V5] mice[13,25] were derived from E3.5 blastocysts using the previously published protocol[7]. TSCs were maintained in defined TX medium[80] comprising DMEM/F12 (ThermoFisher), 1% Pen/Strep (ThermoFisher), 2 mM L-glutamine (ThermoFisher), 64 mg/l L-ascorbic acid 2-phosphate magnesium (Sigma), 543 mg/l NaHCO₃ (Sigma), 14 mg/l sodium selenite (Sigma), 19.4 mg/l insulin (Sigma). 10.7 mg/l holo-transferrin (Sigma), 25 ng/ml human recombinant FGF4 (Sigma), 2 ng/ml TGF-β1 (Peprotech), and 1ug/ml heparin (Sigma) were added prior to use. Cells were cultured at 37 °C and 5% CO₂ on Matrigel-coated dishes and passaged when 80–90% confluent using 0.05% Trypsin (Gibco). Where indicated, Eomes[deg/deg] TSCs were treated by addition of 200 nM dTAG-13

(Tocris), 10 uM BRM014 (MedChemExpress) or, as a control, DMSO alone to the TX culture medium.

## CUT&RUN-seq and data analysis

CUT&RUN was performed as previously described[81]. Briefly, 500,000 Eomes$^{V5/V5}$ TS cells were incubated in wash buffer (20 mM HEPES pH7.5, 150 mM NaCl, 0.5 mM spermidine, containing Roche Complete protease inhibitor EDTA-free) with activated ConA magnetic beads (Bangs Laboratories, BP531). Eomes antibody (abcam, ab2335) or rabbit IgG control (Antibodies online, ABIN101961) in antibody buffer (wash buffer with 0.02% Digitonin, 1 mM EDTA) was added to the bead-bound cells and incubated overnight at 4 °C. Beads were washed with digitonin buffer (wash buffer with 0.02% Digitonin) before incubation with pAG-MNase (Cell Signalling Technology, 40366) in digitonin buffer for 1 h at 4 °C, followed by two washes in digitonin buffer. Chromatin was digested by the addition of 2 mM CaCl2 in digitonin buffer and incubation for 30 min at 0 °C. Chromatin was released by the addition of STOP buffer (340 mM NaCl, 20 mM EDTA, 4 mM EGTA, 0.02% Digitonin, 100 µg/ml RNAse A, 50 µg/ml Glycogen with Spike-in DNA (CST, 40366)) and incubated at 37 °C for 30 min. Following Proteinase K treatment, DNA was phenol-chloroform extracted and resuspended in 40 µl of 1 mM Tris-HCl pH8, 0.1 mM EDTA. CUT&RUN libraries were made using the NEBNext Ultra II DNA Library Prep Kit for Illumina (NEB, E7645) and NEBNext Adaptor/Multiplex Oligos (NEB, E7335S). Libraries were pooled and paired-end sequencing was performed using the Illumina NextSeq 500/550 platform.

Paired-end sequencing fastq files were aligned to the mouse mm10 genome using Bowtie2 version 2.4.4[82], the resulting sam files were converted to bam files, sorted, mitochondrial reads removed with SAMtools version 1.14[83], and files downsampled to contain the same number of reads between test and control files using PICARD version 2.27.4 (https://broadinstitute.github.io/picard/). Peaks were called using MACS2 version 2.2.7.1[84].

Publicly-available ChIP-seq datasets from[28–30] were downloaded from the NCBI Gene Expression Omnibus database (GEO accession codes: GSM3019281; GSM3019325; GSM3040287; GSM3019318; GSM3019290; GSM3019292) and the ArrayExpress Archive of Functional Genomics Data[85] (accession code: E-MTAB-3565). Where necessary, genome coordinates in peak files were converted from mm9 to GRCm38 (mm10) using the UCSC Genome Browser LiftOver tool[86].

## Cell and embryo immunofluorescence (IF) staining

For IF imaging of TSCs, cells were plated onto Matrigel-coated coverslips and fixed in 4% paraformaldehyde (PFA) for 20 min at RT. For embryo imaging, intact decidua were dissected at E5.5 and fixed in 1% PFA in PBS overnight at 4 °C, washed in PBS, embedded in 3% agarose in PBS, manually trimmed and 100 mm sections cut using a Leica VT1000 S vibratome. Coverslips or sections were washed 3 times in PBS (Gibco) containing 0.1% Triton-X (PBS-T), permeabilised for 15 min in 0.5% PBS-T (except for Fgfr1 staining, which didn't receive a permeabilization step), and washed 3 times in 0.1% PBS-T. Samples were blocked in 5% donkey serum/0.2% BSA (Sigma) in 0.1% PBS-T (blocking solution) for 1-2 h at RT, incubated overnight at 4 °C with primary antibody in blocking solution, washed 5 times in PBS-T and fluorophore-conjugated secondary antibodies were added in blocking solution for 2 h at RT. Samples were washed 5 times with 0.1% PBS-T and DAPI in PBS-T was added for 10 min. Samples were washed 3 times in PBS-T, mounted in Vectashield with DAPI (VectorLabs) and imaged on an Olympus Fluoview FV1000 microscope. Image data were processed using ImageJ software. 18 control (Eomes positive) and 4 Eomes null embryos in the decidua were analysed for this study. Antibodies are listed in Supplemental Table 1.

## Western blot

Cells were lysed in RIPA buffer [50 mM Tris pH8.0, 150 mM NaCl, 1% Igepal, 0.5% sodium deoxycholate, 0.1% Sodium dodecyl sulfate (SDS)]. Samples were denatured at 98 °C for 10 min in Laemmli sample buffer (BioRad) containing 10% β- mercaptoethanol, run at 90 V on a Mini-Protean® PAGE gel (BioRad), and transferred onto PVDF membrane for 75 min at 90 V. The membrane was rinsed with distilled H2O, washed in Tris-buffered saline- Tween20 (TBST) (0.1%) for 10 min, blocked with EveryBlot blocking buffer (BioRad) for 10 min at RT and incubated in primary antibody on a shaking platform at 4 °C overnight. The membrane was washed with TBST, incubated in secondary antibody in blocking buffer, washed in TBST, ECL prime (Cytiva) added per the manufacturer's instructions, and exposed to X- ray film. When necessary, the membrane was stripped with stripping buffer (2.9 g glycine, 20 mL SDS in 2 L H2O, pH 2.2), blocked, washed, and exposed to antibody as described above.

## ATAC-seq and data analysis

Tagmentation and indexing of single-cell suspensions ($6 × 10^4$/sample) was performed as previously described[87]. Libraries were purified with a double-sided SPRI AMPure XP bead cleanup (Agencourt) (0.6x to 1.5x bead-to-sample volume ratio). Library size and concentration were determined using the 2100 Bioanalyzer High Sensitivity DNA Kit (Agilent). Samples were sequenced with 75-cycle paired-end Nextera kit with custom Nextera index primers taken from Supplementary Table 1 in Reference[86]. on the Illumina NextSeq 500/550 platform.

Paired-end reads were aligned to the mm10 mouse genome assembly using Bowtie2[82] with the very sensitive option. They were then sorted, mitochondrial reads were discarded using SAMtools[83], and duplicate reads removed using Picard (https://broadinstitute. github.io/picard/). BigWig files were generated using deepTools version 3.5.1[88]. Biological replicates were randomly downsampled to contain the same number of reads for each individual sample, and peaks were called using MACS2 version 2.2.7.1[84]. MACS2-called peaks with a $P$ value of $<10^{-3}$ were used in downstream analyses. Notable changes in ATAC-seq datasets were identified using the DiffBind package on the Galaxy server (www.usegalaxy.org)[89], using bam files and a bed file of all identified peaks in each sample. Genomic Regions Enrichment Analysis Tool (GREAT) 4.0.4 analysis[90] was performed using default parameters with the whole genome as a background to identify peak-gene associations. Enrichment of transcription factor motifs in differentially-accessible peaks was performed using Hypergeometric Optimisation of Motif EnRichment (HOMER)[91]. Heatmaps to assess changes in chromatin accessibility were generated using deepTools version 3.5.1[88].

## RNA isolation and RNA-seq analysis

Total RNA was harvested using a Quick-RNA MicroPrep kit (Zymo) per the manufacturer's instructions. Samples were quantified using Qubit 4.0 Fluorometer (Life Technologies, Carlsbad, CA, USA) and RNA integrity checked with RNA Kit on Agilent TapeStation 4200 (Agilent Technologies, Palo Alto, CA, USA). Library preparation was carried out using NEBNext rRNA Depletion Kit (Human/Mouse/Rat) and NEBNext Ultra II Directional RNA Library Prep Kit for Illumina following manufacturer's instructions (NEB, Ipswich, MA, USA). Briefly, rRNA was depleted with NEBNext rRNA Depletion Kit (Human/Mouse/Rat). rRNA depleted RNAs were fragmented. First strand and second strand cDNA were subsequently synthesised. The second strand of cDNA was marked by incorporating dUTP during the synthesis. cDNA fragments were adenylated at 3' ends, and indexed adapter was ligated to cDNA fragments. Limited cycle PCR was used for library amplification. The dUTP incorporated into the cDNA of the second strand enabled its specific degradation to maintain strand specificity. Sequencing libraries were validated using DNA Kit on the Agilent 5600 Fragment

Analyzer (Agilent Technologies, Palo Alto, CA, USA), and quantified by using Qubit 4.0 Fluorometer (Invitrogen, Carlsbad, CA).

Libraries were multiplexed and sequenced using a 2 × 150 Pair-End (PE) configuration v1.5 on the Illumina NovaSeq 6000 instrument. Image analysis and base calling were conducted using the NovaSeq Control Software v1.7. Raw sequence data (.bcl files) was converted into fastq files and de-multiplexed using Illumina bcl2fastq program version 2.20. One mismatch was allowed for index sequence identification. Paired-end reads were aligned against the mm10 genome using STAR RNA-seq aligner package, with default parameters and −outSAMtype BAM SortedByCoordinate. BAM file primary alignments with mapping quality of >254 were treated as RNA-seq data and imported into SeqMonk 1.48.0 (https://www.bioinformatics.babraham.ac.uk/projects/download.html#seqmonk). The RNA-seq quantitation pipeline using DESeq2[92] was used to identify significantly differentially expressed genes ($P < 0.05$).

### Statistical analysis

Statistical analysis and graph generation was carried out in GraphPad Prism (version 10.0.2). Graphs show mean and standard deviation unless otherwise stated.

### Reporting summary

Further information on research design is available in the Nature Portfolio Reporting Summary linked to this article.

## Data availability

The sequencing data generated in this study has been deposited in the NCBI GEO database, the RNA-seq under the accession code GSE276261, the ATAC-seq under the accession code GSE276039 and the CUT&RUN under the accession code GSE276042. The source data underlying Fig. 1a and b are provided in Supplementary Data 1 and 2 and can also be accessed from the MassIVE repository using the link: ftp://massive-ftp.ucsd.edu/v09/MSV000097887/ (also accessible via proteomeXchange accession code PXD063922).

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

## Acknowledgements

We would like to thank Dr Alan Wainman and the Dunn School Bioimaging Facility, Dr Marjorie Fournier of the University of Oxford Biochemistry Advanced Proteomics Facility and Tim Rostron from the MRC WIMM Sequencing Facility (supported by the MRC TIDU and by the EPA fund CF268) for providing sequencing services. This work was supported by grants from the Wellcome Trust (214175/Z/18/Z E.J.R., 219978/Z/19/Z A.M.B.). E.J.R. is a Wellcome Trust Principal Fellow.

## Author contributions

A.M.B., M.-E.X., I.C., and E.J.R. conceived and performed the experiments, A.M.B. and I.C. performed data analysis, and A.M.B., M.-E.X., I.C., E.K.B., and E.J.R. contributed to writing the paper.

## Competing interests

The authors declare no competing interests.
