## [Transparent Peer Review file · Nature Communications]

Eomesodermin in conjunction with the BAF complex promotes expansion and invasion of the trophoblast lineage

Corresponding Author: Professor Elizabeth J. Robertson

Version 0:

Reviewer comments:

Reviewer #1

(Remarks to the Author)

The manuscript of Bisia and colleagues investigates the function of Eomes in the murine trophoblast lineage. Using trophoblast stem cells (TSCs) as a model system, the Authors found that Eomes partners with the BAF complex, regulating chromatin accessibility and trophoblast gene expression, thereby promoting TSCs maintenance. The study takes advantage of the elegant degron-mediated Eomes protein depletion system previously established in the same lab. I have a few suggestions and questions for the in vitro experiments, which are overall well-designed and executed. The in vivo part can be further extended to better complement the findings in TSC.

1. A general note about the GO analysis (Fig 1e, 2c, 3g, 3h, 4b, S2d, S2e, S4a, S4b). To improve transparency, I suggest including supplementary tables with all GO terms and the genes in these categories. It is important to show what are the actual GO terms, not only to select the ones related to the narrative of trophoblast development. For instance, Fig1e shows only two categories. As a reader, I would like to have a broader view (e.g. the top 10 and I would like to hear the thoughts of the authors about these gene categories). Supplementary tables can also improve the data representations of Fig. 4c and Fig. 4d.

2. For the Discussion part. How does the current work correspond to the findings of Chiara M. Schröder et al., "An EOMES induced epigenetic deflection initiates lineage commitment at mammalian gastrulation"?

3. Cut and Run Eomes data.

Major: How does Eomes genome occupancy correlate to H3K27ac and H3K4me1 (active and active/poised enhancers)?

Minor: Fig1h, the Eomes peak is closer to other genes than Hand1. Moreover, it seems to be positioned in the locus of a gene that is not visible on the graph. Hand1 is upregulated upon Eomes depletion in TSCs, but it may not be directly regulated.

Minor: There are examples of Eomes co-binding (Fig 1g-1i) for all factors in Fig1f except Esrrb.

4. Fig. S1. What are the effects of Brg1-inhibition on TSCs? Does Brg1 inhibition induce a similar phenotype as Eomes depletion? Are the effects concordant with Eomes depletion? What is the timing of the effects, and how this corresponds to the scRNA-seq / ATAC-seq data points?

Fig. S1C. The signal of Fgfr1 should be localised on the cell membrane. This seems not to be the case in Figure S1C. Please provide close-up images or another method for analysing Fgfr1 levels (e.g. Western blot) and pErk levels.

5. Figure 3D. Please include the untreated samples, in addition to 6, 12, 24 and 48h.

6. Figure 3C. Please provide a similar analysis including DMSO and Brg1 inhibition, as well as DMSO, sTAG13 and Brg1 inhibition.

7. BRGi results in a rapid loss of chromatin accessibility as early as 1 hr. It has been noted that after 1 hr of inhibition, there are 28210 less accessible sites (Suppl. Fig. S2c) and around 3000 downregulated genes. Can the authors specify whether

the TSC-associated genes are already affected at this time point?

8. Authors have referred to Goolam, M. et al (2020), regarding Eomes expression in invading TGCs, where it is described as strongly retained. However, the TGC marked in Fig. 5a (arrow) shows very low or absent Eomes expression compared to the ExE. As these studies come from the same lab, it will be nice to clarify whether Eomes is indeed retained or downregulated in the TGCs. Currently, Figure 5a shows clear downregulation of Eomes in the TGCs, compared to the levels in the ExE and the visceral endoderm.

9. The in vitro analyses are performed in TSCs, which represent the ExE in vitro. Thus, a more suitable in vivo system will be the early post-implantation embryo. Moreover, the same lab has elegantly shown that Eomes depletion is possible in the egg cylinder using the degron system. Using this approach the Authors can confirm the downregulation of genes related to trophoblast maintenance (e.g. Sox2, Elf5) and upregulation of genes associated with differentiation (e.g. Hand1, Cited2). In addition, the Authors can also treat E5.5 embryos with Brg1 inhibitor and complement the requested analysis in TSCs.

10. The Authors should also highlight and discuss the effects of Eomes depletion in the pre-implantation embryo (where TE differentiation is arrested, Cdx2 remains uniformly expressed and the TE fail to differentiate), compared to the effects in TSCs and potentially the ExE (where stem cell genes are downregulated and differentiation is initiated). Essentially, these are opposite effects (which makes the proposed analysis at E5.5 ExE lineage more relevant to the presented TSCs analysis in vitro).

11. One of the main statements in the title that Eomes in conjunction with BAF complex promotes invasion is not substantiated.

Clearly Eomes function is necessary for TE to TGCs differentiation as previously shown. However, the proposed regulation of ECM proteins, E-cad and Actg1 do not provide much deeper understanding into the mechanism.

Moreover the in vitro experiments (in TSCs) show the activation of Hand1 and other differentiation markers, which in contrast to the block of differentiation and developmental arrest of Eomes ko blastocysts.

Importantly, there is no evidence that BAF is related to trophoblast invasion (it could be but the manuscript does not experimentally address this question and does not provide direct evidence). None of in vitro experiment using TSCs or the in vivo analysis of Eomes ko embryos in Fig5 directly addresses the role of BAF in the process of invasion.

12. The other statement in the title that Eomes / BAF are essential for the expansion of the trophoblast is unclear. What do the Authors mean by expansion? Is it proliferation? So far the functional experiments in Eomes deficient TSC show upregulation of differentiation genes, indicating role in opposing differentiation / maintenance of the stem cell properties. The effects in vivo, in the trophoblast stem cell pool (ExE) still remain to be determined.

13. Additional minor comments.

- The introduction refers to E2.5 (morula) as a blastocyst stage (line 51)

- TSC can be isolated from post-implantation embryos, but if I remember correctly, the referred papers show E6.5, not E5.5 (lines 66 and 67)

- Figure 1a – a higher resolution image is recommended

Reviewer #2

(Remarks to the Author)

In this paper, the authors investigate the mechanisms by which EOMES regulates trophoblast maintenance, differentiation, and function, primarily using trophoblast stem cells. They describe a hitherto unknown interaction with chromatin remodelling enzymes by studying factors that associate with the same chromatin regions as EOMES and thereby identify a number of trophoblast-associated transcription factors, as well as components of the chromatin remodelling complexes NuRD and BAF. They find that depletion of EOMES and inhibition of BAF both lead to rapid changes in chromatin accessibility, with most EOMES-regulated regions also being regulated by BAF. The authors make use of a rapid depletion system to show that a number of factors regulating cytoskeleton, cell-cell interactions and ECM interaction, as well as trophoblast maintenance and differentiation, change expression at an RNA level within 6 hours of depleting EOMES. They then conclude that EOMES plays a critical role in maintaining chromatin accessibility and therefore the global transcriptional status of trophoblast cells, and more specifically genes critical to TSC identity and function.

Overall, the work is interesting and extends our knowledge of the regulation of trophoblast cells. Additionally, the work has generated several datasets that will be of interest to the community and will facilitate further research. However, the timepoints analysed in this work weaken the ability to draw direct mechanistic links between EOMES and the regulation of the chromatin landscape, in turn somewhat limiting the conclusions that can be drawn from this study.

Fig 2 shows that accessibility is reduced at many regions predicted to regulate TS gene networks following EOMES depletion. Figure 1D, 1F-I, and 2D show that many of these regions are also bound by other TS-associated TFs, while Fig 3 shows that many of these TFs, as well as signal receivers and transducers, change transcript levels within 6 hours. This suggests that many factors will have changed at the protein level within 12 hours and will therefore affect the chromatin landscape. As a result, it is not possible to conclude that EOMES is directly regulating chromatin accessibility. Comments such as "EOMES in conjunction with BRG1 appears to be playing a 'doorstop' role in the sustained maintenance of TE and TB-associated loci of the genome" overstate these results.

Minor comments:

Fig 1C: The majority of EOMES binding sites are far from promoters, so these will likely contribute to a large proportion of the effect, but it may be worth noting that this data actually shows a huge enrichment of EOMES proximal to TSSs (~10-fold higher peaks per kb at a distance of +/- 5kb from TSS vs +/- 50-500kb from TSS), indicating a major selective pressure for EOMES regulation at promoters.

Fig 3B: it is unclear what is the relevance of a 1.5-fold cutoff, since this is not used elsewhere in the RNA-seq analysis. Please could the authors elaborate?

Fig 3C: The effect of density (or perhaps DMSO) on transcription is very strong. As the analysis has been performed against matching timepoints this should not interfere with the results that are presented. However, it would be helpful to see what are the primary changes over time in the DMSO control samples. For example, if there is an increase in differentiation-associated genes, then this could imply that EOMES depletion accelerates rather than induces differentiation of TSCs, or biases their trajectory. A loading plot showing several genes with the strongest contribution to the first two PCs could help with this.

Fig 3E: Please make it clearer which genes are and are not differentially expressed.

Fig 4D: This could benefit from an analysis of the genes with positive vs negative correlation between dTAG and BRG-inhibitor treatment. Clearly some genes show negative correlation (notably including Gata3 and Cited1, which are incorrectly described in the text as showing positive correlation). While there is no reason to expect perfect correlation between these treatments, there could be more depth to the discussion of how the downstream consequences are similar and different.

Figure 5: Please add the number of control and knockout embryos examined and the number with/without evidence of implantation or migration of cells through the endometrial epithelium. A note should be made that this appears to be consistent with the images in Strumpf et al. 2005 [PMID 15788452] and Arnold et al. 2008 [PMID 18171685], though this is the first time it has been explicitly investigated and reported, providing there is sufficient evidence in terms of number of embryos assessed in the current study.

Reviewer #3

(Remarks to the Author)

Version 1:

Reviewer comments:

Reviewer #1

(Remarks to the Author)

I thank the Authors for addressing my questions. I congratulate them on their excellent work and I recommend the manuscript for publication.

I have a few comments on the Authors' responses that can be taken into consideration for the final manuscript. I would leave this decision to the Authors (except number 4 which has not been fully addressed in the main text and must be included in the Discussion).

1. Figure S1C – as I suggested in the original comments, higher magnification images of the membrane localisation of Fgfr1 would be beneficial.

2. Panel A PCA and Panel B PCA provided in the Authors' response could be considered for inclusion in the supplementary data. I think that it makes sense to illustrate the effects of Brg1 inhibition for transparency.

3. The Authors stated that in vivo experiments are technically challenging. According to the experiments in TSCs, a 24-hour time frame should be sufficient to deplete Eomes and observe the effects of the depletion. A 24-hour culture of E5.5 embryos is not technically challenging and there are published protocols. Unfortunately, there was no attempt to perform such experiments. I don't agree that this is out of the scope of the study, the Authors perform in vivo experiments (Figure 5), and have the technical skills and access to the mouse model.

4. In the original comments I raised the following:

The Authors should also highlight and discuss the effects of Eomes depletion in the pre-implantation embryo (where TE differentiation is arrested, Cdx2 remains uniformly expressed and the TE fail to differentiate), compared to the effects in TSCs and potentially the ExE (where stem cell genes are downregulated and differentiation is initiated). Essentially, these are opposite effects.

This has not been addressed, the opposing context-dependent effects have not been mentioned at all. This is important and will require a separate paragraph in the Discussion.

Reviewer #2

(Remarks to the Author)

We thank the authors for their detailed response to all reviewers. We are very happy with the revised manuscript and support the authors' response regarding point 9 from reviewer 1, that in vivo experiments with treatments would be unreasonable and beyond the scope of this paper.

Reviewer #3

(Remarks to the Author)

Response to reviewers

We thank the reviewers for their thoughtful assessment of our manuscript. We were pleased to see the mostly strongly positive feedback and appreciate the opportunity to submit this revised version in which we have addressed the Reviewers comments and concerns, as summarized below. Collectively our new analyses strengthen our conclusions about the role played by Eomes during development of the trophoblast lineage. Hopefully our revised manuscript can now be viewed as acceptable for publication in Nature Communications.

Response to Reviewers Comments

Reviewer #1 (Remarks to the Author):

The manuscript of Bisia and colleagues investigates the function of Eomes in the murine trophoblast lineage. Using trophoblast stem cells (TSCs) as a model system, the Authors found that Eomes partners with the BAF complex, regulating chromatin accessibility and trophoblast gene expression, thereby promoting TSCs maintenance. The study takes advantage of the elegant degron-mediated Eomes protein depletion system previously established in the same lab. I have a few suggestions and questions for the in vitro experiments, which are overall well-designed and executed. The in vivo part can be further extended to better complement the findings in TSC.

1.A general note about the GO analysis (Fig 1e, 2c, 3g, 3h, 4b, S2d, S2e, S4a, S4b). To improve transparency, I suggest including supplementary tables with all GO terms and the genes in these categories. It is important to show what are the actual GO terms, not only to select the ones related to the narrative of trophoblast development. For instance, Fig1e shows only two categories. As a reader, I would like to have a broader view (e.g. the top 10 and I would like to hear the thoughts of the authors about these gene categories). Supplementary tables can also improve the data representations of Fig. 4c and Fig. 4d.

In response to this suggestion, we have generated a new supplementary file showing all GO terms as requested (Supplementary File 3). We highlighted a sub-set of these GO terms in the Figures rather than include the entire set of GO terms as these tended to be very broad and in some cases refer to biological processes we felt were not relevant in the specific context of the cells being examined in our study (such as “abnormal zigzag hair morphology”). Moreover in the case of global perturbation by Brg1 small molecule inhibition, the suite of genes affected is so broad that GO terms span a huge variety of biological processes.

The reviewer also requested inclusion of Supplementary tables for genes reported in Fig. 4c/4d. We note that these were already provided in the original Supplementary Tables, specifically the differentially-expressed genes after 48h of Eomes depletion and BRG1 inhibition respectively (Now Supplemental File 4). We have now summarised the 2503 overlapping genes in a separate sheet of Supp. File 4. In the Figures we elected to use a Venn diagram to summarise the extent of the overlap between these gene lists.

2. For the Discussion part. How does the current work correspond to the findings of Chiara M. Schröder et al., “An EOMES induced epigenetic deflection initiates lineage commitment at mammalian gastrulation”?

Since our original submission the Biorxiv paper mentioned above has now been published under a revised title “EOMES establishes mesoderm and endoderm differentiation potential through SWI/SNF-mediated global enhancer remodeling”. The original Biorxiv manuscript examined interactions in an over-expression experiment carried out in HEK293 cells, rather than an endogenous context which is why we did not originally reference the paper. Interestingly as for our manuscript, this substantially revised paper, now includes both RIME as well as experiments analysing the effects of Brg1 inhibition. We have now cited this paper in the Discussion (lines 412-414, 417-418).

3. Cut and Run Eomes data.

Major: How does Eomes genome occupancy correlate to H3K27ac and H3K4me1 (active and active/poised enhancers)?

We agree this is an interesting issue. As with the ATAC data, we see an overlap of the Eomes binding regions with H3K27Ac and H3K4me1. In summary, H3K27ac peaks (Lee et al., 2019 data) overlap with Eomes CUT&RUN (broad) peakset (of 4145 peaks) within 500 bp of each other: $980/4145 = 23.6\%$ of Eomes C&R peaks.

Above: the 980 sites of Eomes C&R peaks within 500 bp of H3K27ac peaks. Note the CUT&RUN signal is significantly lower than that of the histone mark.

The H3K4me1 peaks (Lee et al., 2019 dataset) overlap with Eomes CUT&RUN (broad) peakset (of 4145 peaks) within 500 bp of each other: $1425/4145 = 34.4\%$ of Eomes C&R peaks.

Above: the 1425 sites of Eomes C&R peaks within 500 bp of H3K4me1 peaks. Note the CUT&RUN signal is significantly lower than that of the histone mark.

Additionally, between the 980 and 1425 Eomes C&R peaks near ac or me1 marks, there is an overlap of 314. Therefore $980 + 1425 - 314 = 2091$.

$2091/4145 = 50.4\%$ of Eomes C&R peaks are within 500 bp of an active/poised enhancer mark.

Minor: Fig1h, the Eomes peak is closer to other genes than Hand1. Moreover, it seems to be positioned in the locus of a gene that is not visible on the graph. Hand1 is upregulated upon Eomes depletion in TSCs, but it may not be directly regulated.

We apologise for the confusion. Hand1 is called as the nearest gene to the Eomes peak but all other transcription units located in this region remain unannotated, including the partially-visible “gene” in that track. For this reason Hand1 was called as the closest gene.

Minor: There are examples of Eomes co-binding (Fig 1g-1i) for all factors in Fig1f except Esrrb.

We thank the Reviewer for pointing this out. We have revised Fig1. g-i to include the Esrrb binding sites.

4. Fig. S1. What are the effects of Brg1-inhibition on TSCs? Does Brg1 inhibition induce a similar phenotype as Eomes depletion? Are the effects concordant with Eomes depletion? What is the timing of the effects, and how this corresponds to the scRNA-seq / ATAC-seq data points?

As expected Brg1 inhibition has a much broader impact on TSCs than that of Eomes depletion including a cessation of cell division as evidenced by cell counts (absolute number) at each timepoint, relative to the starting point (0h):

Bultman et al. (2000) showed that Brg1-null mutant blastocysts do not form functional trophectoderm, as they fail to hatch or attach and form outgrowths *in vitro*, similar to the phenotype of Eomes-null blastocysts. However because both Brg1 null fibroblasts and ESCs can be established (Bultman et al., 2000 Mol Cell; Ho et al. 2011, Nat Cell Biol) Brg1 is not a general cell viability factor. Rather its function appears to be essential during early embryo development, and most likely its loss of function selectively affects formation and function of the trophectoderm lineage.

Fig. S1C. The signal of Fgfr1 should be localised on the cell membrane. This seems not to be the case in Figure S1C. Please provide close-up images or another method for analysing Fgfr1 levels (e.g. Western blot) and pErk levels.

Apologies about this oversight. Our original staining was performed on PFA fixed permeabilised cells. To reveal membrane bound Fgfr1 we have now repeated this experiment using permeabilization free PFA fixed conditions enabling us to visualise both membrane bound and intracellular Fgfr1. These new images are now included in revised Supplementary Figure S1C. together with higher magnification images as requested to visualise membrane and intracellular Fgfr1.

5. Figure 3D. Please include the untreated samples, in addition to 6, 12, 24 and 48h.

Unfortunately, it is not possible to include 0h/starting timepoint as each column represents an intra-timepoint comparison between dTAG-treated and DMSO/control-treated conditions. As 0h only had one condition (untreated) it isn't possible to replicate this type of analysis.

6. Figure 3C. Please provide a similar analysis including DMSO and Brg1 inhibition, as well as DMSO, sTAG13 and Brg1 inhibition.

We performed this analysis, as shown below. However, the PCA plot including BRG1 inhibition samples becomes less informative as it obscures the trajectory of the control (DMSO-treated) cells. This is due to magnitude of the effect of BRGi relative to control

treatment or even dTAG treatment/Eomes depletion (Panel A). The plot with BRGi and control cells alone (Panel B) does highlight the trajectory of both BRGi and control cells across time, but the major axis still corresponds to the transcriptional changes of BRGi (PC1). Accordingly we don't believe these additional analyses adds any new insight to the manuscript.

Panel A

Panel B

7. BRGi results in a rapid loss of chromatin accessibility as early as 1 hr. It has been noted that after 1 hr of inhibition, there are 28210 less accessible sites (Suppl. Fig. S2c) and around 3000 downregulated genes. Can the authors specify whether the TSC-associated genes are already affected at this time point?

1h after BRGi, 634 genes are differentially expressed in TSCs. A few trophoblast-associated terms are highlighted in GO analysis, including “trophoblast giant cell differentiation” (binomial fold enrichment = 10.34; raw P-val = 9.09E-5 and encompassing the genes *Prdm1*, *Elf5*, *E2f8*, *Erf*, *Lif*). The other is “placenta blood vessel development” (binomial fold

enrichment = 5.89; raw P-val = 4.88E-04 and encompassing the genes *Syde1*, *Plcd3*, *Fosl1*, *Hes1*, *Arid1a*, *Socs3*). *Eomes* and *Cited2* are also misregulated at this timepoint. Thus, a handful of TSC-related genes and genes associated with downstream placental development are already mis-regulated 1 hour after inhibition. We have added a sentence to this effect in the revised manuscript (line 236-237).

8. Authors have referred to Goolam, M. et al (2020), regarding *Eomes* expression in invading TGCs, where it is described as strongly retained. However, the TGC marked in Fig. 5a (arrow) shows very low or absent *Eomes* expression compared to the ExE. As these studies come from the same lab, it will be nice to clarify whether *Eomes* is indeed retained or downregulated in the TGCs. Currently, Figure 5a shows clear downregulation of *Eomes* in the TGCs, compared to the levels in the ExE and the visceral endoderm.

To clarify this point and highlight the *Eomes* staining in the various embryonic populations we have revised Figure 5 to include *Eomes*-only staining panels for the wildtype embryo. Differences between the image in Goolam et al (2020) and the image shown here can be attributed to technical differences - we used IHC in Goolam (2020), whereas here we used IF imaging which better represents the relative expression levels. Additionally, polyploid TGCs contain an enlarged nucleus, making the IF *Eomes* signal appear weaker. Finally, it's interesting that *Eomes* levels in the visceral endoderm are also substantially lower compared to the ExE. Nonetheless we have shown that *Eomes* plays an essential role in the visceral endoderm (Nowotschin et al 2013, *Genes & Dev*). *Eomes* performs its critical roles despite these possibly different expression levels.

9. The *in vitro* analyses are performed in TSCs, which represent the ExE *in vitro*. Thus, a more suitable *in vivo* system will be the early post-implantation embryo. Moreover, the same lab has elegantly shown that *Eomes* depletion is possible in the egg cylinder using the degron system. Using this approach the Authors can confirm the downregulation of genes related to trophoblast maintenance (e.g. *Sox2*, *Elf5*) and upregulation of genes associated with differentiation (e.g. *Hand1*, *Cited2*).

In addition, the Authors can also treat E5.5 embryos with Brg1 inhibitor and complement the requested analysis in TSCs.

We agree that investigating the role of *Eomes* at varying timepoints during implantation and trophoderm development would be an interesting extension of the current studies. However, there are substantial technical barriers to performing these experiments. To date, embryo culture protocols have focused on using gastrulation stage embryos as the starting material which develop normally for 24-48 hours (Rivera-Perez et al., 2010; McDole et al., 2018). By contrast culturing E5.0/E5.5 embryos has proven challenging. Although a polar trophoderm culture invasion protocol has been published by the Hiiragi lab (Ichikawa et al., 2022 *Dev Cell*) there are no established methods for intact E5.5 embryos which would be needed to study development and expansion of the early ExE. We previously showed that transient culture of E6.0 embryos in the presence of d-TAG results in *Eomes* depletion (Bisia et al 2023) but have been unable to devise protocols which allow these embryos to develop further *ex vivo*. These experiments are, unfortunately, beyond the scope of the current paper. These limitations are why originally chose the *in vitro* TSC system to model *Eomes* functions during implantation/early post-implantation development.

The proposed Brg1 inhibition experiments are also technically problematic as Brg1 inhibition has a very profound effect on development of the trophectoderm including causing growth arrest of TSCs. Even if robust embryo culture conditions could be established, it is unclear if Brg1 inhibition of E5.0-5.5 embryos would simply arrest their development. It seems unlikely that these experiments would enhance the current in vitro findings.

10. The Authors should also highlight and discuss the effects of Eomes depletion in the pre-implantation embryo (where TE differentiation is arrested, Cdx2 remains uniformly expressed and the TE fail to differentiate), compared to the effects in TSCs and potentially the ExE (where stem cell genes are downregulated and differentiation is initiated). Essentially, these are opposite effects (which makes the proposed analysis at E5.5 ExE lineage more relevant to the presented TSCs analysis in vitro).

Eomes appears to have distinct functions in the divergent trophoblast sub-populations present at the blastocyst stage. It is initially required in the mural TE which plays an essential role for entosis and migration into the uterine stroma and which can be visualized within the adjacent stroma as individual giant cells (which retain Eomes expression). We show in Figure 5 that Eomes deficient blastocysts, elicit a decidual response (expected since a drop of oil or glass bead also cause a decidual response) but fail to implant appropriately since the mural TE fails to breach the luminal epithelium.

Slightly later in post-implantation stage embryos, the polar TE overlying the ICM expands to give rise to the ExE, strongly expressing Eomes. Subsequent down-regulated expression of Eomes is associated with the emergence of the most proximal differentiated (Hand1 positive) cells of the ectoplacental cone – progenitors of the outer spongiotrophoblast layer of the placenta.

We have discussed Eomes functions in the mural versus polar TE in the final paragraphs of the Discussion. To help further clarify this point we have added an additional sentence to the discussion (line 472-475).

11. One of the main statements in the title that Eomes in conjunction with BAF complex promotes invasion is not substantiated.

Clearly Eomes function is necessary for TE to TGCs differentiation as previously shown. However, the proposed regulation of ECM proteins, E-cad and Actg1 do not provide much deeper understanding into the mechanism.

Moreover the in vitro experiments (in TSCs) show the activation of Hand1 and other differentiation markers, which in contrast to the block of differentiation and developmental arrest of Eomes ko blastocysts.

Importantly, there is no evidence that BAF is related to trophoblast invasion (it could be but the manuscript does not experimentally address this question and does not provide direct evidence). None of in vitro experiment using TSCs or the in vivo analysis of Eomes ko embryos in Fig5 directly addresses the role of BAF in the process of invasion.

Unfortunately, “TSCs” are not a homogenous population. Rather they represent a very heterogenous population of self-renewing cells derived over a period of several weeks/months following embryo dispersal. As shown in Supp Fig 1C Eomes and Tfap2c (TSC

markers) staining show varying expression levels in TSC cultures maintained in Tx 4++++ defined culture medium suggesting the population likely contains both mural and polar derivatives. This explains why on the one hand simple “blastoids” can be formed by mixing TSC and ESC derived EPSCs together (Sozen et al, 2019, Dev Cell), while longer term culture of mixtures of TSCs, ESCs and XEN cells gives rise to “synthetic” embryos, where the TSC derived “ExE” enables patterning of the ESC-derived “epiblast” like structure (Harrison et al., 2017, Science).

Because of this TSC heterogeneity, our experiments have uncovered a potential role for Eomes in the mural TE for maintaining expression of genes such as E-Cad and Actg1 which have roles in cellular architecture and invasion. As TSC also mimic the behaviour of the polar TE derived ExE population, our experiments also identified genes such as Hand1 and other markers which are up-regulated in the outer differentiated cells of the EPC in response to down-regulated Eomes expression.

It remains unknown as to whether BAF complexes are required for TE invasion and migration. The Brg1 null mutant embryos arrest at implantation but whether or not they implant correctly via mural TE invasion has not been examined. However, a large cohort of the ECM genes are misregulated on Brg1 inhibition, as shown in Figure 4. Therefore as both Eomes and Brg1 control the expression of these genes, suggests that Brg1 may also play a role in the invasion of the uterine epithelium by the mural TE derived TGCs. This remains an area for future investigation, since it would require the generation of Brg1-degron tagged TSCs.

12. The other statement in the title that Eomes / BAF are essential for the expansion of the trophoblast is unclear. What do the Authors mean by expansion? Is it proliferation? So far the functional experiments in Eomes deficient TSC show upregulation of differentiation genes, indicating role in opposing differentiation / maintenance of the stem cell properties. The effects in vivo, in the trophoblast stem cell pool (ExE) still remain to be determined.

We used the term “expansion” to refer both to cell proliferation (as in the expansion of the early undifferentiated ExE progenitor population) as well as the expansion and diversification of the TE lineage during post-implantation development. The polar TE normally expands following implantation to form the ExE, which contains the TSC pool giving rise to terminally differentiated derivatives in the EPC. The polar TE fails to expand or proliferate in Eomes-null mutants, supporting our hypothesis that Eomes affects TE expansion. Unfortunately, as mentioned previously, we are unable to validate this in vivo at E5.5 given the lack of suitable culture protocols.

13. Additional minor comments.

- The introduction refers to E2.5 (morula) as a blastocyst stage (line 51)
- TSC can be isolated from post-implantation embryos, but if I remember correctly, the referred papers show E6.5, not E5.5 (lines 66 and 67)
- Figure 1a – a higher resolution image is recommended

We thank the Reviewer for pointing out these errors which have been corrected in the revised manuscript. We have also included a higher resolution image of the volcano plot in Figure 1a.

Reviewer #2 (Remarks to the Author):

In this paper, the authors investigate the mechanisms by which EOMES regulates trophoblast maintenance, differentiation, and function, primarily using trophoblast stem cells. They describe a hitherto unknown interaction with chromatin remodelling enzymes by studying factors that associate with the same chromatin regions as EOMES and thereby identify a number of trophoblast-associated transcription factors, as well as components of the chromatin remodelling complexes NuRD and BAF. They find that depletion of EOMES and inhibition of BAF both lead to rapid changes in chromatin accessibility, with most EOMES-regulated regions also being regulated by BAF. The authors make use of a rapid depletion system to show that a number of factors regulating cytoskeleton, cell-cell interactions and ECM interaction, as well as trophoblast maintenance and differentiation, change expression at an RNA level within 6 hours of depleting EOMES. They then conclude that EOMES plays a critical role in maintaining chromatin accessibility and therefore the global transcriptional status of trophoblast cells, and more specifically genes critical to TSC identity and function.

Overall, the work is interesting and extends our knowledge of the regulation of trophoblast cells. Additionally, the work has generated several datasets that will be of interest to the community and will facilitate further research. However, the timepoints analysed in this work weaken the ability to draw direct mechanistic links between EOMES and the regulation of the chromatin landscape, in turn somewhat limiting the conclusions that can be drawn from this study.

Fig 2 shows that accessibility is reduced at many regions predicted to regulate TS gene networks following EOMES depletion. Figure 1D, 1F-I, and 2D show that many of these regions are also bound by other TS-associated TFs, while Fig 3 shows that many of these TFs, as well as signal receivers and transducers, change transcript levels within 6 hours. This suggests that many factors will have changed at the protein level within 12 hours and will therefore affect the chromatin landscape. As a result, it is not possible to conclude that EOMES is directly regulating chromatin accessibility. Comments such as “EOMES in conjunction with BRG1 appears to be playing a ‘doorstop’ role in the sustained maintenance of TE and TB-associated loci of the genome” overstate these results.

We were pleased to see these very positive comments and appreciate the Reviewers concerns about our interpretation of the data.

Further support for our hypothesis that Eomes may play a “doorstop” role in the TE is provided by a very recent paper just published in Dev Cell showing Eomes, through its interaction with SWI/SNF, binds to enhancers of mesoderm and endoderm genes prior to transcription initiation (Schroder et al. 2024, EOMES establishes mesoderm and endoderm differentiation potential through SWI/SNF-mediated global enhancer remodeling). This work, carried out independently of our studies, and similarly utilizing RIME and Brg1i approaches, concludes that Eomes plays a very similar role during mesoderm and endoderm

specification to that we propose for the TE lineage. We have now cited this work in the discussion (line 412-414, 417-418)

In Response to the Reviewers queries about timing/accessibility and expression levels. We see modest changes at the transcript level of a sub-set of the TFs and chromatin regulators highlighted as Eomes binding partners in Fig 1D-I and also as motifs in differentially accessible regions in Fig 2D, 6 hours following Eomes depletion. These genes include: Elf5 (0.15 Log2FC), Tfp2c (-0.399 Log2FC), Pou3f1 (0.398 Log2FC), Sox2 (0.14 Log2FC) and Meis1 (0.209 Log2FC), as shown in Supplemental File 3. By contrast expression levels of other genes including Zfp281, Lsd1, Esrrb or Tead4 are not detectably different after 6 hours. However, differences in expression become more pronounced by 12 hours. Since Eomes is the most highly down-regulated component of the network after 12 hours, we suggest that loss of Eomes likely accounts for the majority of the observed accessibility changes.

Additionally, the TSC cell population is not homogenous as revealed by Eomes or Tfp2c staining (as shown in Supp Figure S1) indicative of a fairly heterogeneous set of cell states. This phenomenon has been well documented in TSCs (Kubaczka et al., 2014, Stem Cell Reports; Perez-Garcia et al., 2021, Elife; Seong et al., 2022, Cell Stem Cell). As the ATAC-seq was performed on bulk populations and not at a single cell level, there is intrinsic noise in differential analysis of the data that may be obscured at early stage.

Collectively for these reasons we conclude that the differentially accessible sites observed at 12 hours are, on balance, most likely contributed by Eomes loss rather than indirectly via downstream consequences. Consistent with this suggestion, analysis of Eomes null mutant embryos reveals a more profound and earlier phenotype in comparison to loss of other components, for example Elf5 and Tfp2c, which have defects in later development (Donnison et al., 2005, Development; Auman et al., 2002, Development).

Minor comments:

Fig 1C: The majority of EOMES binding sites are far from promoters, so these will likely contribute to a large proportion of the effect, but it may be worth noting that this data actually shows a huge enrichment of EOMES proximal to TSSs (~10-fold higher peaks per kb at a distance of +/- 5kb from TSS vs +/- 50-500kb from TSS), indicating a major selective pressure for EOMES regulation at promoters.

The standard way of presenting the distribution of peak distances from TSSs is via 0-5kb, 5-50kb, 50-500kb "binning". Peaks per kb is not a commonly used way of describing peak distribution. Furthermore, in addition to the absolute majority of peaks being >50kb from the nearest TSS (Figure 1c), 50.4% of Eomes peaks are also within 500 bp of an active/poised enhancer mark, further reinforcing the idea that Eomes is binding to and regulating the accessibility status of thousands of regions in the TSC genome (please see response to Reviewer 1, comment 3).

Fig 3B: it is unclear what is the relevance of a 1.5-fold cutoff, since this is not used elsewhere in the RNA-seq analysis. Please could the authors elaborate?

We elected to use a >1.5-fold cutoff to highlight the fact that although in absolute numbers Eomes loss results in up- and down-regulation of similar numbers of genes, the magnitude of the effect is greater for genes downregulated upon Eomes loss. Thus, Eomes appears to promote expression of a larger number of genes than those it represses.

Fig 3C: The effect of density (or perhaps DMSO) on transcription is very strong. As the analysis has been performed against matching timepoints this should not interfere with the results that are presented. However, it would be helpful to see what are the primary changes over time in the DMSO control samples. For example, if there is an increase in differentiation-associated genes, then this could imply that EOMES depletion accelerates rather than induces differentiation of TSCs, or biases their trajectory. A loading plot showing several genes with the strongest contribution to the first two PCs could help with this.

We agree that cell density has a marked effect on the transcriptional profiles. When designing the RIME experiment we compared cell density, days post-splitting and Eomes staining to ensure that our bulk TSC cultures exhibited the highest levels of Eomes+ cells. We noticed that the staining was the most heterogeneous as the cultures approach confluence while they are the most uniform in both morphological appearance and Eomes staining 24 hours post splitting. However as they expand the cultures form shallow craters. The larger flatter more differentiated (Eomes^{low}) cell population become situated centrally while the edges are comprised of smaller Eomes^{high} cells. However when split and re-seeded all colonies are composed of Eomes^{high} small cells.

Because thousands of genes contribute to each of the first two PCs (on the order of 10,000), we feel that a loading plot would not be very informative as it would by necessity represent a very narrow, cherry-picked sub-section this large cohort. We have therefore selected the top 1000 genes contributing most highly to each extreme of each of the two PCs and carried out a GO analysis on each of the 4 gene lists. For the Reviewers information we have now included this analysis as an additional File (uploaded as an xlsx file "PCA GO terms").

Briefly, PC1 genes with lowest rotations (contributing more to early timepoints) are associated with "cellular response to TGF- β stimulus", "regulation of epithelial cell proliferation", "negative regulation of epithelial cell differentiation", "epithelial morphogenesis", "in utero embryonic development", and "regulation of stress fibre assembly". GO analysis of PC1 genes with highest rotations (contributing more to late timepoints) gives terms associated with cytoskeleton and ECM organization, cell migration and adhesion, and Wnt and Ras signalling pathways.

PC2 genes with lowest rotations (contributing more to DMSO-treated control TSCs) are associated with "negative regulation of cell differentiation involved in embryonic placental development", "sequestering of TGF- β in extracellular matrix", "maintenance of epithelial apical/basal polarity", "positive regulation of MAPKKK cascade by FGFR signaling pathway", "positive regulation of stem cell proliferation", and terms associated with cell-cell adhesion, cytoskeletal, and motility regulation. Finally, PC2 genes with highest rotations (contributing more to Eomes-depleted TSCs) are associated with cytoskeletal regulation, cell adhesion, and ECM terms, "embryonic placenta development", "collagen-activated tyrosine kinase receptor signaling pathway", "PDGFR α/β signaling pathway", "integrin-mediated signaling

pathway”, “positive regulation of blood vessel endothelial cell migration”, “negative regulation of cell population proliferation”, and “positive regulation of cell differentiation”.

Fig 3E: Please make it clearer which genes are and are not differentially expressed.

The Fig. 3e legend has been updated and clarified accordingly.

Fig 4D: This could benefit from an analysis of the genes with positive vs negative correlation between dTAG and BRG-inhibitor treatment. Clearly some genes show negative correlation (notably including Gata3 and Cited1, which are incorrectly described in the text as showing positive correlation). While there is no reason to expect perfect correlation between these treatments, there could be more depth to the discussion of how the downstream consequences are similar and different.

We thank the Reviewer for pointing this out. In the revised text we have removed the incorrectly-described genes and as requested we have also expanded on this point in the Discussion (lines 449-455).

Figure 5: Please add the number of control and knockout embryos examined and the number with/without evidence of implantation or migration of cells through the endometrial epithelium. A note should be made that this appears to be consistent with the images in Strumpf et al. 2005 [PMID 15788452] and Arnold et al. 2008 [PMID18171685], though this is the first time it has been explicitly investigated and reported, providing there is sufficient evidence in terms of number of embryos assessed in the current study.

Apologies for this oversight, we analysed 18 control (Eomes positive) and 4 Eomes null embryos in the decidua. None of the 4 Eomes null embryos showed any evidence of implantation, as judged by lack of invasion through the uterine epithelium. We have now included these numbers in the revised manuscript (Line 618-619).

It is well known that a decidual response can be induced in rodents via an intrauterine injection of oil droplets, as well as other foreign materials, e.g. polysaccharides (Finn CA & Keen PM (1963), J. Embryol. Exp. Morphol., Andrale CGTJ et al., (1996) The Anatomical Record, Loeb L (1908) JAMA). Hence decidualisation is not equivalent to implantation. Although Eomes null blastocysts induce a decidual response, as we show here the embryo does not implant but rather remains in the uterine lumen. The mutant mural TE derived TGC cells cannot invade into the maternal decidua.

Strumpf et al 2005, showed images of the pre-implantation Eomes null embryos and a single image of a presumptive Eomes null blastocyst like structure within the decidua. We only showed blastocysts and *in vitro* blastocyst outgrowth cultures in Arnold et al 2008. However, Russ et al., 2000 (Nature) showed low resolution H&E stained sections of e6.5 decidua. A second image showed a LacZ positive Eomes null embryo within the decidua that has not expanded. Unfortunately, it is impossible to evaluate invasion in these H&E sections. For this reason we used the paternally inherited mTmG allele to fully visualise all embryo derived cells within the decidua. We have previously shown a Otx2 stained Eomes null embryo (Bisia et al., 2023 PNAS) that has not expanded past the blastocyst stage, similar to the data

presented in Strumpf (2005) and Russ (2000). Thus while historically Eomes has been described as having a “peri-implantation” lethal phenotype, the current study unambiguously shows that while Eomes null embryos initiate a decidual response, the embryos fail to implant due to defects in TE invasion.

Reviewer #3 (Remarks to the Author):
